# Compelling ReLU Networks to Exhibit Exponentially Many Linear Regions at Initialization and During Training

## Abstract

A neural network with ReLU activations may be viewed as a composition of piecewise linear functions. For such networks, the number of distinct linear regions expressed over the input domain has the potential to scale exponentially with depth, but it is not expected to do so when the initial parameters are chosen randomly. Therefore, randomly initialized models are often unnecessarily large, even when approximating simple functions. To address this issue, we introduce a novel training strategy: we first reparameterize the network weights in a manner that forces the network to exhibit a number of linear regions exponential in depth. Training first on our derived parameters provides an initial solution that can later be refined by directly updating the underlying model weights. This approach allows us to learn approximations of convex, one-dimensional functions that are several orders of magnitude more accurate than their randomly initialized counterparts. We further demonstrate how to extend our approach to multidimensional and non-convex functions, with similar benefits observed.

## 1 Introduction

Beyond complementary advances in areas like hardware, storage, and networking, the success of neural networks is primarily due to their ability to efficiently capture and represent nonlinear functions (Gibou et al., 2019). In a neural network, the goal of an activation function is to introduce nonlinearity between the network's layers so that the network does not simplify to a single linear function. The rectified linear unit (ReLU) has a unique interpretation in this regard. Since it either deactivates a neuron or acts as an identity, the resulting transformation on each individual input remains linear. However, each possible configuration of active and inactive neurons can produce a unique linear transformation over a particular region of input space. The number of these activation patterns and their corresponding linear regions provides a way to measure the expressivity of a ReLU network[1] and can theoretically scale exponentially with the depth of the network (Montufar et al., 2014a; Serra et al., 2018). Hence deep architectures may outperform shallow ones.

Surprisingly, though, a sophisticated theory of how to best encode functions into ReLU networks is lacking, and in practice, adding depth is often observed to help less than one might expect from this exponential intuition. Lacking more advanced theory, practitioners typically use random parameter initialization and gradient descent, the drawbacks of which often lead to extremely inefficient solutions. Hanin & Rolnick (2019) show a rather disappointing bound pertaining to randomly initialized networks: they prove that the average number of linear regions formed upon initialization is entirely independent of the configuration of the neurons, so depth is not properly utilized. They observed that gradient descent has a difficult time creating new activation regions and that their bounds approximately held after training. As we will discuss later, the number of linear regions is not actually a model property that gradient descent can directly optimize. Gradient descent is also prone to redundancy; Frankle & Carbin (2019) show how around 95% of weights may ultimately be eliminated from a network without significantly degrading accuracy.

The present work aims to begin eliminating these inefficiencies, starting in a simple one-dimensional setting. Drawing inspiration from existing theoretical ReLU constructions, our novel contributions

---

[1]See Appendix A.2 for definitions of terms like linear regions, activation patterns, and activation regions.

include a special reparameterization of a ReLU network that forces it to maintain an exponential number of activation patterns over the input domain. We then demonstrate a novel pretraining strategy, which trains these derived parameters before manipulating the underlying matrix weights. This allows the network to discover solutions that are more accurate and unlikely to be found otherwise. Our technique directly rectifies the issues raised by Hanin & Rolnick (2019). Firstly, we initialize immediately with an exponential number of linear regions, which would not be expected to happen otherwise. Secondly, we are not reliant on gradient descent to "discover" new regions; we only need to maintain the existing ones, which is already guaranteed by the reparameterization during the pretraining stage. Additionally, this pretraining step can avoid areas of the loss landscape where unassisted gradient descent might make short-sighted optimizations that eliminate activation regions. Our results demonstrate that minimizing the reliance of network training on unassisted gradient descent can reliably produce error values orders of magnitude lower than a traditionally-trained network of equal size. Although preliminary numerical demonstrations in this theoretical study pertain to relatively low-dimensional functions, the paper concludes with our views on extending these theoretical exponential benefits for ReLU networks to arbitrary smooth functions with arbitrary dimensionality, which would have significant practical utility.

## 2 RELATED WORK

This work is primarily concerned with a novel training methodology, but it also possesses a significant approximation theoretic component. In one dimension, where we initially focus, our work is a generalization of the approximation to $x^2$ we review in this section. The reparameterization we employ modifies this method to become trainable to represent other convex one-dimensional functions, and then converts that result back into a matrix representation. The goal of this method is not to re-express the full set of neural network weights, but to constrain them into efficient patterns.

### 2.1 FUNCTION APPROXIMATION

Infinitely wide neural networks are known to be universal function approximators, even with only one hidden layer (Hornik et al., 1989; Cybenko, 1989). Infinitely deep networks of fixed width are universal approximators as well (Lu et al., 2017; Hanin, 2019). In finite cases, one may study trade-offs between width and depth to assess a network's ability to approximate (learn) a function.

Notably, there exist functions that can be represented with a sub-exponential number of neurons in a *deep* architecture, yet which require an exponential number of neurons in a *wide and shallow* architecture. For example, Telgarsky (2015) shows that deep neural networks with ReLU activations on a one-dimensional input are able to generate symmetric triangle waves with an exponential number of linear segments (shown in Figure 1 as the ReLU network $T(x)$).

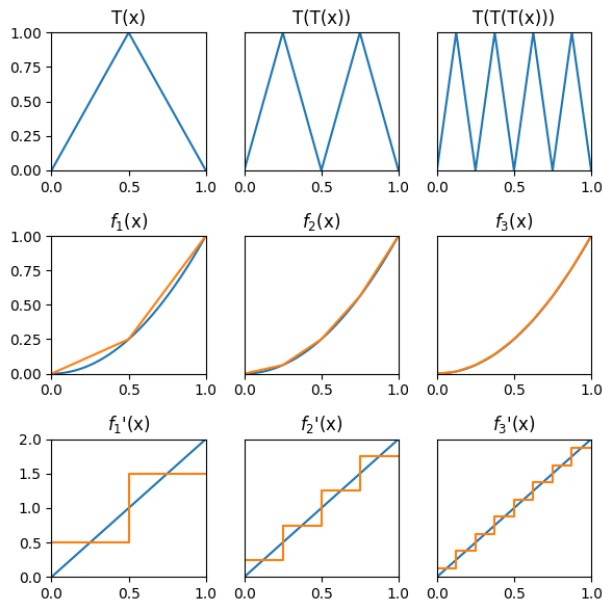

Figure 1: (Top to bottom) Composed triangle waves; using collections of the above function to approximate $x^2$; derivatives of the above approximations.

This network functions as follows: each layer takes a one-dimensional input on $[0, 1]$, and outputs a one-dimensional signal also on $[0, 1]$. The function they produce in isolation is a single symmetric triangle. Together in a network, each layer inputs its output to the next, performing function composition. Since each layer converts lines from 0 to 1 into triangles, it doubles the number of linear segments in its input signal, exponentially scaling with depth.

The same effect can be achieved with non-symmetric triangle waves (Huchette et al., 2023) (or any shape that sufficiently "folds" the input space (Montufar et al., 2014b)). Our reparameterization strategy focuses on non-symmetric triangle waves. The location on $(0, 1)$ of a triangular function's peak acts as an alternative parameterization of the function, instead of its weight matrix representation. Unlike the raw weights of a depth $d$ ReLU network, any values chosen for the triangle peak parameters will result in the creation of $2^d$ linear regions. Furthermore, this parameterization is trainable. By using the peak location to set the raw weights of a layer, the gradients can backpropagate through the raw weights to update the triangle peaks, effectively confining the network to a subspace of weights that generates many linear regions.

The dilated triangular waveforms produced in this manner are not particularly useful on their own. Their oscillations quickly become excessively rapid, and their derviatives do not exist everywhere (especially in the infinite depth limit). But these problems can be rectified by taking a sum over the layers of a network. Yarotsky (2017) and Liang & Srikant (2016) construct $y = x^2$ on $[0, 1]$ with exponential accuracy using symmetric triangle waves. To produce their approximation, one begins with $f_0(x) = x$, then computes $f_1(x) = f_0(x) - T(x)/4$, $f_2(x) = f_1(x) - T(T(x))/16$, $f_3(x) = f_2(x) - T(T(T(x)))/64$, and so forth, as pictured in Figure 1. As these successive approximations are computed, Figure 1 plots their convergence to $x^2$, as well as the convergence of the derivative to $2x$. Our reparameterization generalizes this approximation to use non-symmetric triangle waves to approximate a wider class of convex, differentiable, one-dimensional functions.

The $x^2$ approximation is used by other theoretical works as a building block to guarantee exponential convergence rates in more complex systems. Perekrestenko et al. (2018) construct a multiplication gate via the identity $(x + y)^2 = x^2 + y^2 + 2xy$. The squared terms can all be moved to one side, expressing the product $xy$ as a linear combination of squared terms. They then further assemble these multiplication gates into an interpolating polynomial, which can have an exponentially decreasing error when the interpolation points are chosen to be the Chebyshev nodes. Polynomial interpolation does not scale well into high dimensions, so this and papers with similar approaches will usually come with restrictions that limit function complexity: Wang et al. (2018) requires low input dimension, Montanelli et al. (2020) uses band limiting, and Chen et al. (2019) approximates low-dimensional manifolds. These works all make use of a fixed representation of $x^2$. If our networks were substituted in for the $x^2$ approximation, these works would provide theoretical guarantees about the capabilities of the resulting model. Even though the approximation rates will not scale well with input dimension, they serve as a bound that can be improved upon. In Section 5 we further elaborate on how to use our networks to represent higher-dimensional or non-convex functions.

Other works focus on showing how ReLU networks can encode and subsequently surpass traditional approximation methods (Lu et al., 2021; Daubechies et al., 2022), including spline-type methods (Eckle & Schmidt-Hieber, 2019). Interestingly, certain fundamental themes from above like composition, triangles, or squaring are still present. Another interesting comparison of the present work is to Ivanova & Kubat (1995), which uses decision trees as a means to initialize sigmoid neural networks for classification. Similar to the spirit of our work, which restricts parameterizations of ReLU networks, Elbrächter et al. (2019) explores theoretical aspects of the conditionining of ReLU network training and provides constructive results for a parameterization space that is well-conditioned. Chen & Ge (2024) present a creative approach where they explore reparameterizing the direction of weight vectors using hyperspherical coordinates to improve training dynamics. Unlike their reparameterization, ours will restrict the network's expressivity in order to prevent it from learning inefficient weight patterns. Lastly, Park et al. (2021) approaches the problem of linear region maximization from an information theory perspective and uses a loss penalty rather than a reparameterization to increase the number of linear regions.

## 2.2 Neural Network Initialization

Our work seeks to improve network initialization by making use of explicit theoretical constructs. This stands in sharp contrast the current standard approach, which treats neurons homogeneously. Two popular initialization methods implemented in PyTorch are the Kaiming (He et al., 2015) and Xavier initialization (Glorot & Bengio, 2010). They use weight values that are sampled from distributions defined by the input and output dimension of each layer. Aside from sub-optimal approximation power associated with random weights, a common issue is that the initial weights and biases in a ReLU network can cause every neuron in a particular layer to output a negative value.

The ReLU activation then sets the output of that layer to 0, blocking any gradient updates. This is referred to as the dying ReLU phenomenon (Qi et al., 2024; Nag et al., 2023). Worryingly, as depth goes to infinity, the dying ReLU phenomenon becomes increasingly likely (Lu et al., 2020). Several papers propose solutions: Shin & Karniadakis (2020) use a data-dependent initialization, while Singh & Sreejith (2021) introduce an alternate weight distribution called RAAI that can reduce the likelihood of the issue and increase training speed. We observed during our experiments that RAAI greatly reduces, but does not eliminate the likelihood of dying ReLU. Our approach enforces a specific network structure that does not collapse in this manner.

# 3 INITIALIZATION AND PRETRAINING CONSTRUCTION

We begin by discussing how to deliberately architect the weights of a 4-neuron-wide, depth $d$ ReLU network to induce a number of linear segments exponential in $d$. As alluded to in Section 2.1, our initialization algorithm is simply to associate a nonsymmetric triangle function with each layer, and then to randomize the location of each triangle's peak in $(0, 1)$; any choice produces $2^d$ regions. This yields a reparameterization of the ReLU network in terms of the triangle peak locations, $a_i$, rather than in terms of the network's weights. The pretraining step of our algorithm then involves learning and updating these parameters. This is done by setting individual neuron weights in accordance with the procedures described in this section, and then backpropagating the gradients to update the triangle peak parameters. Pseudocode for our entire algorithm is provided in Appendix A.1.

To illustrate how we set weights, we first describe the mathematical functions that arise in our analysis. Triangle functions are defined as

$$T_i(x) = \begin{cases} \frac{x}{a_i} & 0 \leq x \leq a_i \\ 1 - \frac{x - a_i}{1 - a_i} & a_i \leq x \leq 1 \end{cases}$$

where $0 < a_i < 1$. This produces a triangular shape with a peak at $x = a_i$ and with both endpoints satisfying $y = 0$. Each layer considered in isolation would compute these if directly fed the input signal. $T_i(x)$'s derivatives are the piecewise linear functions:

$$T_i'(x) = \begin{cases} \frac{1}{a_i} & 0 < x < a_i \\ \frac{1}{1 - a_i} & a_i < x < 1 \end{cases} \quad (1)$$

In a deep network, the layers feed into each other, composing their respective triangle functions:

$$W_i(x) = \bigcirc_{j=0}^{i} T_j(x) = T_i(T_{i-1}(...T_0(x))) \quad (2)$$

These triangle waves will have $2^i$ linear regions, doubling with each layer. The output of the network in pretraining will be a weighted sum over the triangle waves formed at each layer. Assuming the network to be infinitely deep, we have

$$F(x) = \sum_{i=0}^{\infty} s_i W_i(x) \quad (3)$$

where $s_i$ are scaling coefficients on each of the composed triangular waveforms $W_i$.

To encode these functions into the weights of a ReLU network, we begin with triangle functions (see the right subnetwork in Figure 2). Its maximum output is 1 at the peak location $a \in (0, 1)$. Neuron $t_1$ simply preserves the input signal. Meanwhile, $t_2$ is negatively biased, deactivating it for inputs less than $a$. Subtracting $t_2$ from $t_1$ changes the slope at the point where $t_2$ begins outputting a nonzero signal. The weight $-1/(a - a^2) = -(1/a + 1/(1 - a))$ is picked to completely negate $t_1$'s positive influence, and then produce a negative slope. When these components are combined into a deep network, the individual triangles they form will be composed with those earlier in the network, but the two neurons will still behave analogously. Considering the output waveform of $t_1$ and $t_2$ neurons over the entire input domain $[0, 1]$ and at arbitrary depth, we have that $t_1$ neurons are always active, outputting complete triangle waves. In contrast, $t_2$ neurons are deactivated for small inputs, so they output an alternating sequence of triangles and inactive regions (see Figure 3).

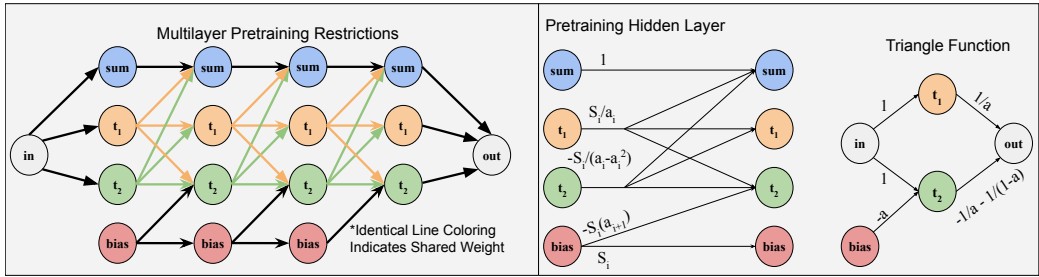

Figure 2: On the right is a network representation of a triangle function. The middle shows that triangle function as a hidden layer of a network. The one-dimensional input and output of a triangle function is converted into shared weights. A full pretraining network is assembled on the left.

Naively using the output of one triangle generator (from Figure 2) as input to the next would form a $1 \times 2 \times 1 \times 2 \times 1...$ shape, but this is unnecessarily deep. We can replicate the one-dimensional function composition in the hidden layers on the left side of Figure 2 by using weight sharing instead. Any outgoing weight from $t_1$ or $t_2$ is shared; every neuron taking in a triangle wave as input does so by combining $t_1$ and $t_2$ in the same proportion. In this way, we can avoid having to use the extra intermediate neurons. This is evident in Figure 3 where the waves output by $t_1$ and $t_2$ neurons posses identical slopes in regions where both neurons are active.

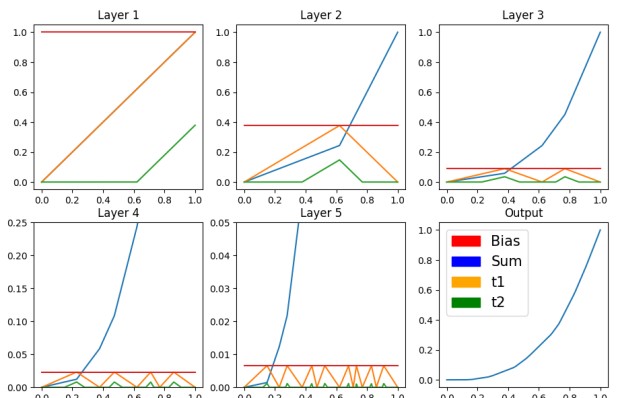

Figure 3: Each colored line shows the output signal of a neuron with respect to the input to the network. Colors match the corresponding neurons in Figure 2.

There are two other neurons in the pretraining network's layers, each with a specific role. The accumulation neuron (marked as "sum" in Figure 2) maintains a weighted sum of all previous triangle waves through each layer. This is similar to a residual connection (He et al., 2016), except that no other neurons take the sum as input (its output is convex, and therefore not rapidly oscillating, and detrimental to making more linear regions). If the sum neuron were naively implemented, it would multiply the $t_1$ and $t_2$ weights by the sum coefficients. Based on Theorem 3.1, these coefficients will be exponentially decaying, so learning these weights directly may cause conditioning issues. Instead, all weights in each layer are multiplied by the ratios between successive scaling coefficients, so that the outputs of $t_1$ and $t_2$ neurons decay in amplitude in each layer. A conventional bias will have no connections to prior layers, so it will be unable to adjust to the rescaling neuron outputs without having to learn exponentially small values. Therefore, a fourth neuron is configured to output a constant signal, so the weight other neurons place on this output can replace their bias. The bias neuron will connect to the previous layer's bias neuron so that the constant signal can scale down gradually with each layer. We handle this scaling explicitly, rather than relying on regularizers like batch norm (Ioffe & Szegedy, 2015) or layer norm (Lei Ba et al., 2016), partially in order to preserve the mathematical transparency of our model, and partially because this solution arises naturally from our derivations.

## 3.1 DIFFERENTIABLE MODEL OUTPUT

Given the rapid oscillations of the triangle waves formed at each layer, the network will output a fractal with many choices of scaling parameters. This would be poorly predictive of unseen data points generated from a smooth curve. Our main mathematical result addresses this issue by forcing differentiability of the network output. To achieve differentiability, it turns out that for this class of

functions, the peak locations of the triangle waves uniquely determine the scales with which to sum them. For a broader mathematical discussion, as well as sufficient conditions for differentiability in the limit, see the appendix.

**Theorem 3.1.** *$F(x)$ is continuously differentiable on $[0, 1]$ only if the scaling coefficients $s_i$ are selected based on the triangle peaks $a_i$ according to:*

$$s_{i+1} = s_i(1 - a_{i+1})a_{i+2} \tag{4}$$

## 4 One-Dimensional Experiments

The goal of deep learning is to train a network to approximate a (generally unknown) nonlinear function. Accordingly, in this section, we implement our method to learn several convex one-dimensional curves. These nonlinear functions are known—so that we can measure error—and we remark and demonstrate results on more difficult functions in Section 5, Section 6, and Appendix A.9. The aim of these experiments is twofold: (1) we would like to determine how to learn the most effective function representations possible, and (2) to explore how the utilization of an increased number of linear regions can affect a network's ability to capture underlying nonlinearity in its training data. To demonstrate that our networks can learn function representations that better utilize depth, we benchmark against PyTorch's (Paszke et al., 2019) default settings (`nn.linear()` uses Kaiming initialization), as well as the RAAI distribution from Singh & Sreejith (2021), and produce errors that are *orders of magnitude lower than both*. For our experimental networks, we use a common set of initialization points where the triangle peaks and scaling parameters are chosen according to our main theorem (Equation 3.1). We compare the effect of reparameterized pretraining against skipping immediately to standard gradient descent (from a reparameterized initialization as in Section 3). We also compare training the scaling parameters freely during pretraining, instead of choosing them to achieve differentiability. The intent of both comparisons is to see if pretraining and further differentiability constraints facilitate smoother navigation of the loss landscape. Lastly, we conduct a second round of tests to determine if pretrained networks display an enhanced predictive capacity on unseen data points, as might be expected if they can leverage greater nonlinearity in their outputs.

### 4.1 Experimental Setup

All models are trained using Adam (Kingma & Ba, 2017) as the optimizer with a learning rate of 0.001 for 1,000 epochs to ensure convergence (see Appendix for a brief analysis of different learning rates). Each network is four neurons wide with five hidden layers, along with a one-dimensional input and output. The loss function used is the mean squared error, and the average and minimum loss are recorded for 30 models of each type. The networks unique to this paper share a common set of starting locations using the construction in Section 3, so that the effects of each training regimen are directly comparable. As in related papers (Perekrestenko et al., 2018; Daubechies et al., 2022), we focus for the moment on one-dimensional examples, which are sufficient to demonstrate our proposed theory and methodology. The four curves we approximate are $x^3$, $x^{11}$, $\tanh(3x)$, and a quarter period of a sine wave. The curves are chosen to capture a variety of convex one-dimensional functions. To approximate the sine and the hyperbolic tangent, the triangle waves are added to the line $y = x$. For the other approximations, the waves are subtracted. This requires the first scaling factor to be $a_0 * a_1$ instead of $(1 - a_0) * a_1$. The first set of data is 500 evenly spaced points on the interval $[0, 1]$ for each of the curves. This is chosen to be very dense deliberately, to try to evoke the most accurate representations from each network. We determine from these tests that pretraining with differentiability enforced produces the best results, so we compare it to standard networks in our second set of experiments. We use a second set of data consisting of only 10 points, with a test set of 10 points spaced in between so as to be as far away from learned data as possible. The goal of this set of experiments is to compare the predictive capacities of the networks on unseen data.

### 4.2 Numerical Results

Our first set of results are shown in Tables 1 and 2, wherein we observe several important trends. First, the worst performing networks are the Default Networks that rely on randomized (Kaiming)

Table 1: Minimum and mean (30 samples) MSE error approximating $y = x^3$ and $x^{11}$.

| Training Type | Min $x^3$ | Min $x^{11}$ | Mean $x^3$ | Mean $x^{11}$ |
|---|---|---|---|---|
| Default Network (Kaiming) | $2.11 \times 10^{-5}$ | $2.19 \times 10^{-5}$ | $7.20 \times 10^{-2}$ | $2.82 \times 10^{-2}$ |
| RAAI Distribution | $2.14 \times 10^{-5}$ | $4.40 \times 10^{-5}$ | $3.97 \times 10^{-2}$ | $4.12 \times 10^{-2}$ |
| Pretraining Skipped | $7.63 \times 10^{-7}$ | $1.86 \times 10^{-5}$ | $3.89 \times 10^{-5}$ | $3.56 \times 10^{-4}$ |
| Differentiability Not Enforced | $1.64 \times 10^{-7}$ | $3.20 \times 10^{-6}$ | $1.02 \times 10^{-5}$ | $3.73 \times 10^{-5}$ |
| Differentiability Enforced | $\mathbf{7.86 \times 10^{-8}}$ | $\mathbf{8.86 \times 10^{-7}}$ | $\mathbf{5.27 \times 10^{-7}}$ | $\mathbf{7.87 \times 10^{-6}}$ |

Table 2: Minimum and mean (30 samples) MSE error approximating $y = \sin(x)$ and $y = \tanh(3x)$.

| Training Type | Min $\sin(x)$ | Min $\tanh(3x)$ | Mean $\sin(x)$ | Mean $\tanh(3x)$ |
|---|---|---|---|---|
| Default Network (Kaiming) | $4.50 \times 10^{-5}$ | $5.75 \times 10^{-5}$ | $1.15 \times 10^{-1}$ | $1.96 \times 10^{-1}$ |
| RAAI Distribution | $3.59 \times 10^{-5}$ | $1.09 \times 10^{-5}$ | $3.63 \times 10^{-2}$ | $2.31 \times 10^{-2}$ |
| Pretraining Skipped | $1.96 \times 10^{-7}$ | $1.07 \times 10^{-6}$ | $1.93 \times 10^{-5}$ | $8.38 \times 10^{-5}$ |
| Differentiability Not Enforced | $\mathbf{4.41 \times 10^{-8}}$ | $1.49 \times 10^{-7}$ | $1.47 \times 10^{-5}$ | $3.81 \times 10^{-4}$ |
| Differentiability Enforced | $5.06 \times 10^{-8}$ | $\mathbf{6.82 \times 10^{-8}}$ | $\mathbf{2.21 \times 10^{-7}}$ | $\mathbf{8.42 \times 10^{-7}}$ |

initialization. Even the networks that forgo pretraining benefit from initializing with many activation regions. When pretraining constraints are used, they are able to steer gradient descent to the best solutions, resulting in reductions in minimum error of three orders of magnitude over default networks. Pretraining with differentiability enforced also closes the gap between the minimum and mean errors compared to other setups. This indicates that these loss landscapes are indeed the most reliable to traverse. Enforcing differentiability during pretraining can impart a bias towards smoother solutions during subsequent unassisted gradient descent.

The last trend to observe is the poor average performance of default networks. In a typical run of these experiments, around half of the default networks collapse. RAAI is able to eliminate most, but not all of the dying ReLU instances due to its probabilistic nature, so it, too, has high mean error.

Our second set of results is shown in Table 3 and in Figure 4. Here the most important impact of utilizing exponentially many linear regions is demonstrated. Not only can more accurate repre-

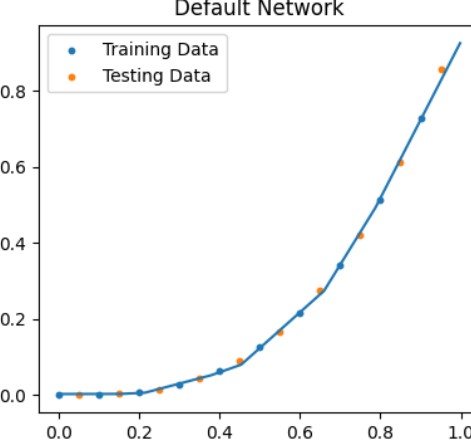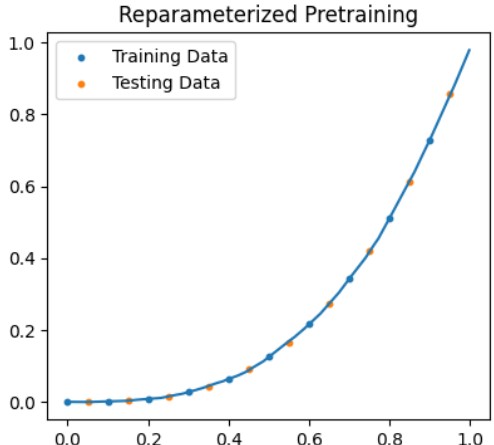

Figure 4: Comparison between standard Kaiming initialization/gradient descent and pretraining with differentiablity enforced. Using more linear regions allows the curve to better predict the test points.

Table 3: Minimum errors on unseen points from training on sparse data.

| Training Type | Min $x^3$ | Min $x^{11}$ | Min $\sin(x)$ | Min $\tanh(3x)$ |
|---|---|---|---|---|
| Default Network (Kaiming) | $2.41 \times 10^{-4}$ | $2.14 \times 10^{-3}$ | $2.27 \times 10^{-5}$ | $1.60 \times 10^{-4}$ |
| Differentiable Pretraining | $\mathbf{5.65 \times 10^{-6}}$ | $\mathbf{6.53 \times 10^{-4}}$ | $\mathbf{7.92 \times 10^{-7}}$ | $\mathbf{5.09 \times 10^{-6}}$ |

sentations of training data be learned, but more linear regions allow the network to better capture underlying nonlinearity to enhance its predictive power in regression tasks. This result is especially significant because it indicates that even in cases where there are fewer data points than linear regions, having the additional regions can still provide performance advantages.

### 4.3 Gradient Descent does not Directly Optimize Efficiency

Figure 5 shows the interior of a default network. The layers here are shown before applying ReLU. The default networks fail to make efficient use of ReLU to produce linear regions, even falling short of 1 bend per neuron, which can easily be attained by forming a linear spline (1 hidden layer) that interpolates some of the data points. Examining the figure, the first two layers are wasted. No neuron's activation pattern crosses $y = 0$, so ReLU is never used. Layer 3 could be formed directly from the input signal. Deeper in the network, more neurons remain either strictly positive or negative. Those that intersect $y = 0$ are monotonic, only able to introduce one bend at a time. The core issue is that while more bends leads to better accuracy, net-

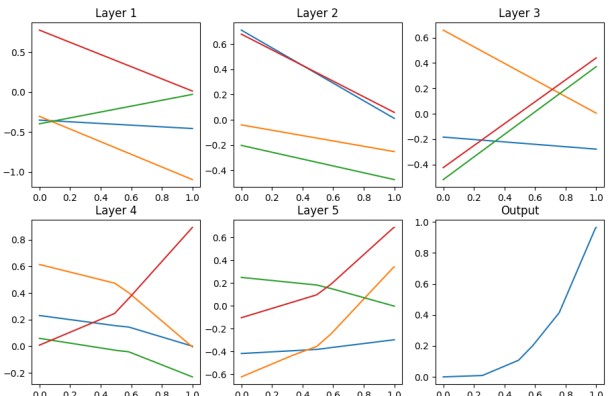

Figure 5: Approximation produced with standard methodologies (shown before ReLU is applied). The neuron colors here are arbitrary; they do not correspond to Figure 2.

works that have few bends are not locally connected in parameter space to those that have many. This is problematic since gradients can only carry information about the effects of infinitesimal parameter modifications. If a bend exists, gradient descent can reposition it. But for a neuron that always outputs a strictly positive value (such as the red in layer 2), bends cannot be introduced by infinitesimal weight or bias adjustments. Therefore, bend-related information will be absent from its gradients. Gradient descent will only compel a network to bend by happenstance; indirectly related local factors must guide a neuron to begin outputting negative values. Occasionally, these local incentives are totally absent, and the network outputs a bend-free line of best fit.

### 4.4 Effect of Reparameterized Initialization vs Pretraining

Figure 6 compares the top performing reparameterized models for $x^3$ with and without pretraining. We observe that without the guidance of the pretraining, gradient descent usually loses the triangle generating structure around layer 4 or 5, devolving into noisy patterns and resulting in higher errors. Pretraining maintains structure at greater depths. This behavior of gradient descent we observe is problematic since theoretical works often rely on specific constructions within networks to prove their results. Gradient descent greedily abandons any such structure during its optimization in favor of models that can be worse in the long term. A theoretical result that shows a certain representation exists in the set of neural networks will thus have a hard time actually learning it without a subsequent plan to control training. This phenomenon highlights the need for theoretical work to derive more expressive training controls, which can give better guidance for the optimization of functions of greater dimension or complexity.

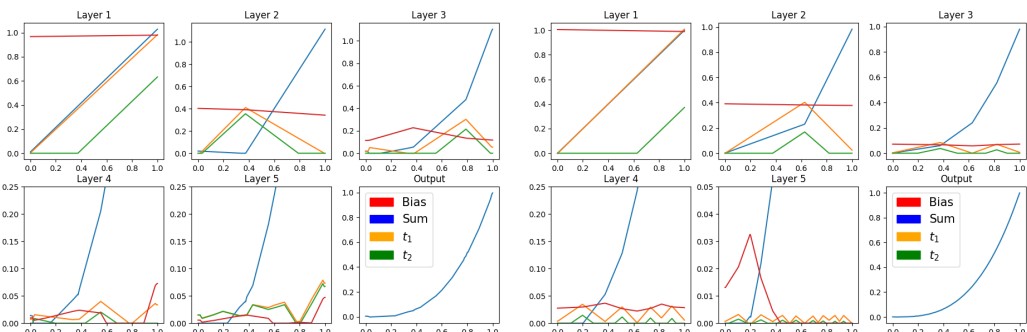

Figure 6: Pretraining (right) results in better structural retention in deeper layers.

# 5 EXTENSION TO NON-CONVEX FUNCTIONS AND HIGHER DIMENSIONS

The results presented so far are limited to one-dimensional convex functions, but these are not limitations of our method. Our constructed networks can be applied to approximate higher-dimensional functions by using these networks as activation functions in a larger network (see Figure 9 in the Appendix for a schematic). For instance, in two dimensions, we take two or more copies of our network and estimate a target function as a linear combination of the networks' outputs. Because the weights of the linear combination are randomized, each copy of the network will tend to learn different lower-dimensional aspects of the target function, whereby the combination can accurately capture the higher-dimensional function.

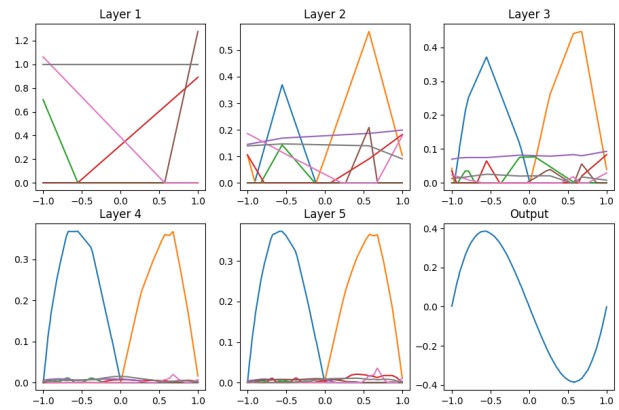

Figure 7: Approximation of $y = x^3 - x$ by difference of pretrained components, achieving a loss of $4 \times 10^{-7}$. A standard $8 \times 5$ network yielded a larger loss of $8 \times 10^{-6}$.

The same approach can address non-convex functions since many non-convex functions (including every continuous function with a bounded second derivative (Hessian)) can be expressed as the difference of two convex functions (see Zhang et al. (2018), which also describes how every ReLU network can be decomposed into a difference of two piecewise linear convex functions). We do not claim that two copies of our network are sufficient for learning every pathological function, but preliminary results in this section are surprisingly promising. In fact, we highly encourage reviewing Appendix A.9, which demonstrates our method on real-world datasets and problems with up to eight input dimensions.

We note that the approach in this section bears similarity to the recently popularized Kolmogorov-Arnold Networks (KAN) (Liu et al., 2024), which use spline functions as activations of a network where every weight is 1. This is based on the Kolomogorov-Arnold representation theorem (Kolmogorov, 1957; Arnold, 1957; 1959), which gives a guarantee that every continuous multivariate function can be represented by a neural network-like structure with one-dimensional activation functions. But by restricting the activation functions to splines, the KAN abandons its theoretical guarantees. Using our networks as activations, however, maintains the theoretical ability to perform polynomial interpolation since they retain the ability to approximate $x^2$, and the multiplication gate discussed in Section 2.1 is based on a linear combination of squared terms.

Figures 7 and 8 show learning the non-convex function $y = x^3 - x$ and the two-dimensional function $z = r^3$ on $[-1, 1]$, each by using two of our convex function blocks. Rather than directly using our

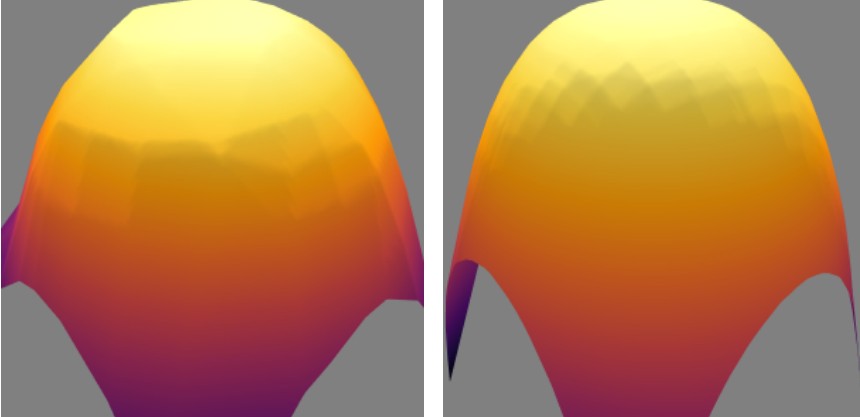

Figure 8: Approximations of $z = \sqrt{x^2 + y^2}^3$ using an $8 \times 5$ regular network (left) and a union of two of our pretrained components (right). Losses are $1.5 \times 10^{-4}$ and $3.5 \times 10^{-6}$, respectively, demonstrating a nearly two orders of magnitude improvement using our techniques.

networks as activation functions, we instead encode each of their layers as a block diagonal structure in one combined matrix (which has the same effect). Two slight modifications are made to each of the component networks to better interact with ReLU. First, two neurons are used for the sum so that a negative component may be maintained in the case of $x^3 - x$, which would otherwise be zeroed out by ReLU. Second, the triangular functions are generated upside-down, preventing ReLU from zeroing the function outside of the approximation region.

For both $x^3 - x$ and $r^3$, we substantially outperform standard initialization and training procedures. Especially striking is the increased number of linear regions in Figure 8. Visible in Figure 7 is the gradient descent-induced destructuring discussed in Section 4.4. This likely occurs due to the block diagonal structure of the larger weight matrices, which results in many zeros for the second pass of gradient descent to fill in unassisted. This could be addressed either by limiting training only to the block diagonal entries, or by deriving a more advanced pretraining method that makes use of the full matrix. Note that since $x^3 - x$ is twice differentiable we can write it as $(x^3 - x + 3x^2) - 3x^2$, a difference of functions convex on $[-1, 1]$ obtained by finding a parabola based on trying to make the derivative of $x^3 - x$ monotone. Note that this is not a unique decomposition.

# 6 CONCLUDING REMARKS

This paper focused on exploiting the potential computational complexity advantages neural networks offer for the problem of efficiently learning nonlinear functions; in particular, compelling ReLU networks to approximate functions with exponential accuracy as network depth is linearly increased. Our results showed improvements of one to several orders of magnitude in using our initialization and pretraining strategy to train ReLU networks to learn various nonlinear functions, including non-convex and multi-dimensional functions. This finding is particularly powerful since random initialization and gradient descent are not likely to produce an efficient solution on their own, even if it can be proven to exist in the set of sufficiently sized ReLU networks. Although we proposed one preliminary strategy for extending our network for higher-dimensional functions, this strategy has limitations, such as leaving many weights to be filled in by gradient descent (which it generally does not do effectively). We anticipate continuing investigating strategies for improving the generalizability of our work, which we believe is critically important since it offers the possibility of an exponentially more accurate drop-in replacement of linear layers in any architecture. We note that there are other metrics for evaluating a neural network's expressivity (e.g., Raghu et al. (2017)), which would be interesting to measure for our networks in future work. Overall, we are hopeful that future works by our group and others will help illuminate a complete theory for harnessing the potential exponential power of depth in ReLU and other classes of neural networks.

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

# A APPENDIX

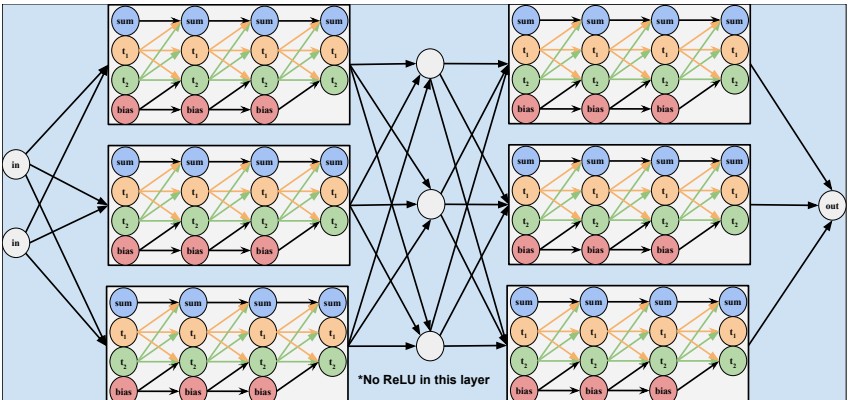

Figure 9: Extending our results into a larger network. ReLU should not be used in the layers between the blocks, as creating nonconvex functions this way will tend to make a lot of negative values.

## A.1 INITIALIZATION AND TRAINING ALGORITHM

The initialization step of our algorithm is to generate a vector $A = [a_0, a_1, ...a_n]^T$, where each $a_i$ is randomized in $(0, 1)$. Given this $A$, the pretraining step of our algorithm sets the weights of the input ($I$), hidden ($H_i, 1 \leq i \leq n-1$), and output ($O$) layers of the network as follows:

$$I(x) = \begin{bmatrix} x \\ x \\ x \\ 0 \end{bmatrix} + \begin{bmatrix} 0 \\ 0 \\ -a_0 \\ 1 \end{bmatrix}$$

$$H_i(x) = \begin{bmatrix} 1 & S_i/a_i & -S_i/(a_i - a_i^2) & 0 \\ 0 & S_i/a_i & -S_i/(a_i - a_i^2) & 0 \\ 0 & S_i/a_i & -S_i/(a_i - a_i^2) & -S_i a_{i+1} \\ 0 & 0 & 0 & S_i \end{bmatrix} \times ReLU(H_{i-1}(x))$$

$$O(x) = \begin{bmatrix} 1 & S_n/a_n & -S_n/(a_n - a_n^2) & 0 \end{bmatrix} \times ReLU(H_{n-1}(x))$$

where $S_i$ can either be chosen independently, or chosen based on $A$. In the latter case, Equation 3.1 gives $S_i = s_i/s_{i-1} = (1 - a_i)a_{i+1}$. This assignment of $I$, $H$, and $O$ is used in each iteration of the pretraining algorithm.

---

**Algorithm 1** Initialization and Pretraining

$A \leftarrow \text{Random}((0, 1)^n)$
**while** Epochs > 0 **do**
    Network $\leftarrow$ Set_Weights($A$)          ▷ Set weights as above each iteration
    Loss $\leftarrow$ (Network($x$) − $y$)$^2$
    Network-Gradient $\leftarrow$ Derivative(Loss, Network)          ▷ Regular Backpropagation
    A-Gradient $\leftarrow$ Derivative(Network, $A$)          ▷ Backpropagate through weight setting
    Gradient $\leftarrow$ Network-Gradient × A-Gradient
    $A \leftarrow A − \epsilon \times$ Gradient          ▷ Update A, Not the network
**end while**

---

The network weights can then be initialized once more based on the learned vector $A$, to then update the weights via regular gradient descent. In our experiments, both phases of training (pretraining and gradient descent) ran for 1000 epochs. Full pseudocode is listed in Algorithm 1.

## A.2 TERMINOLOGY

Before presenting further formal mathematical details of our method, we first briefly review a few pieces of basic ReLU network terminology used in the paper. The reader is referred to the Chmielewski-Anders (2020) for an excellent alternative presentation of these terms.

An *activation pattern* is a boolean mask that tracks which neurons in a network have their output zeroed by ReLU activations. The *activation regions* of a ReLU network are connected (and can be shown to be convex) sets of inputs on which the activation pattern is constant. Since the action of the ReLU activation function is constant, the network output over an activation region is equivalent to the case where there is no activation function and the associated zeroed neurons are absent; therefore, the network output behaves linearly over the inputs in the activation region. Relatedly, a *linear region* is a set of inputs on which the network output behaves linearly with respect to its inputs. It may consist of multiple neighboring activation regions. Another important concept is the *boundary* of a neuron, as described in Rolnick & Kording (2020): the set of inputs for which the neuron outputs 0, independently of ReLU. The boundary of a neuron is precisely the boundary of the activation regions it adds to the network. Chen & Ge (2024) refers to this set as the *characteristic activation boundary* since these are the boundaries of the activation regions.

We note that although works like ours are theoretical papers that leverage these concepts, studying these ideas can lead to interesting applied learning research. For instance, Rolnick & Kording (2020) leverages theoretical works like those cited in our related work discussion for an exciting privacy application: it is proven that a ReLU network's output can often be provably used to reverse engineer the architecture of a ReLU network, up to isomorphism.

## A.3 MATHEMATICAL RESULTS

For convenience, we first restate the functions defined in the main body of the paper.

$$T_i(x) = \begin{cases} \frac{x}{a_i} & 0 \le x \le a_i \\ 1 - \frac{x - a_i}{1 - a_i} & a_i \le x \le 1 \end{cases}$$

$$T_i'(x) = \begin{cases} \frac{1}{a_i} & 0 < x < a_i \\ \frac{1}{1 - a_i} & a_i < x < 1 \end{cases}$$

$$W_i(x) = \bigcirc_{j=0}^{i} T_j(x) = T_i(T_{i-1}(...T_0(x)))$$

$$F(x) = \sum_{i=0}^{\infty} s_i W_i(x)$$

The goal of this section will be to show how to select the $s_i$ based on $a_i$ in a manner where the derivative $F'(x)$ is defined and continuous on all of $[0, 1]$. We begin by assuming that

$$F'(x) = \sum_{i=0}^{\infty} s_i W_i'(x)$$

We will see that the resulting choice of $s_i$ ensures uniform convergence of the derivative terms, so that the derivative of the infinite sum is indeed infinite sum of the derivatives. Fortunately, the left and right derivatives $F_+'(x)$ and $F_-'(x)$ already exist everywhere, since each bend in each $W_i$ (where the full derivative of $F$ is undefined) is assigned the slope of the line segment to its left or right respectively. The $s_i$ scaling values will have to be chosen appropriately so that $F_+'(x)$ and $F_-'(x)$ are equal for all bend points.

Notationally, we will denote the sorted $x$-locations of the peaks and valleys of $W_i(x)$ by the lists $P_i = \{x : W_i(x) = 1\}$ and $V_i = \{x : W_i(x) = 0\}$. We will use the list $B_i$ to reference the locations of all non-differentiable points, which we refer to as bends. $B_i := P_i \cup V_i$.

$f_i(x) = \sum_{n=0}^{i-1} s_n W_n(x)$ will denote finite depth approximations up to but not including layer $i$. The error function $E_i(x) = \sum_{n=i}^{\infty} s_n W_n(x) = F(x) - f_i(x)$ will represent the error between the finite approximation and the infinite depth network. This odd split around layer $i$ makes the proofs cleaner.

Figure 10 highlights some important properties about composing triangle functions. Peaks alternate with valleys. Peak locations in one layer become valleys in the next. Valleys in one layer remain valleys in all future layers since 0 is a fixed point of each $T_i$. To produce $W_i$, each line segment of $W_{i-1}$ becomes a dilated copy of $T_i$. Each triangle function has two distinct slopes $1/a_i$ and $-1/(1 - a_i)$ which are dilated by the chain rule during the composition. On negative slopes of $W_{i-1}$, the input to layer $i$ is reversed, so those copies of $T_i$ are reflected. Due to the reflection, the slopes of $W_i$ on each side of a peak or valley are proportional. Alternatively, one could consider that on each side of a peak in $W_{i-1}$, there is a neighborhood of points that are greater than $a_i$, and are composed with the same line segment of $T_i$ that has slope $-1/(1 - a_i)$. Either way, it's important to note that the slopes on each side of a bend scale identically during each composition.

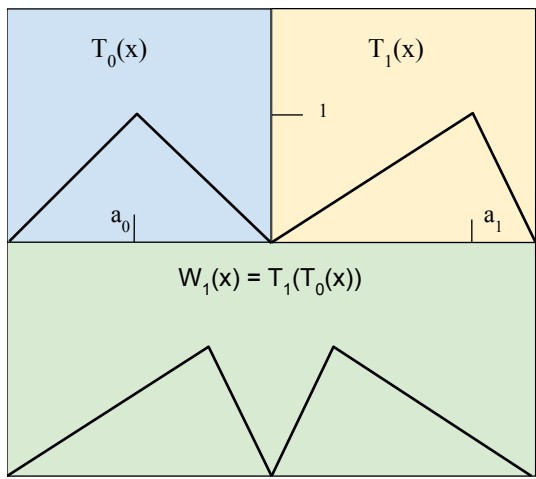

Figure 10: Triangle functions $T_0$ and $T_1$, and the triangle wave resulting from their composition. Note how $T_1$ is reflected in $W_1$.

Before we begin reasoning about $F'(x)$, it can simplify the analysis to only consider the derivative of the error function $E'(x)$.

**Lemma A.1.** *for $x \in P_i$, $F'(x)$ is defined if and only if $E_i'(x)$ is defined.*

*Proof.* All of $W_n(x)$ for $n < i$ are differentiable at $x \in P_i$ since $x$ will lie in the interior of a linear region of $W_n$. Therefore, $f_i'(x) = \sum_{n=0}^{i-1} s_n W_n'(x)$ exists at these points. Since $E_i' + f_i' = F'$, $F'(x)$ is defined if and only if $E_i'(x)$ is defined. $\square$

Thanks to the previous lemma, we only need to work with $E_i$. Here we compute the right derivative $(E_i)_+'(x)$ of the error function at a point $x$. The left derivative will only be different by a constant factor.

**Lemma A.2.** *For all $x \in P_i$, $E_+'(x)$ and $E_-'(x)$ are proportional to*

$$s_i - \frac{1}{1 - a_{i+1}} \left( s_{i+1} + \sum_{n=i+2}^{\infty} s_n \prod_{k=i+2}^{n} \frac{1}{a_k} \right). \tag{5}$$

*Proof.* Let $x_k$ be some point in $P_i$, and let $k$ be its index in any list it appears in. To calculate the value of $E_+'(x_k) = \sum_{n=i}^{\infty} s_n (W_n)_+'(x_k)$, we will have to find the slope of the linear intervals to the immediate right of $x_k$ for all $W_i$. We will use $R_x$ to represent $W_{i+}'(x_k)$. The first term in the sum will be $R_x s_i$. Since the derivatives of composed functions will multiply from the chain rule, so the value of the next term is $W_{i+1}'(x_k) = T_{i+1}'(W_i(x_k)) R_x$. $T_{i+1}$ has two linear segments, giving two slope possibilities to multiply by. The correct one to choose is $-1/(1 - a_{i+1})$ because it's 'active' around $x_k$ ($x_k$ is a peak of $W_i$, so $W_i(x_k) > a_{i+1}$ for $x \in (B_{i+1}[k - 1], B_{i+1}[k + 1])$). This gives $W_{i+1}'(x_k) = -R_x \frac{s_{i+1}}{(1 - a_{i+1})}$. Note that the second term has the opposite sign as the first.

For all remaining terms, since $x_k$ was in $P_i$, it is in $V_j$ for $j > i$. For $x \in (B_{j+1}[k - 1], B_{j+1}[k + 1])$, $W_j(x) < a_{j+1}$ and the chain rule applies the slope $1/a_{j+1}$. Since this slope is positive, every

remaining term continues to have the opposite sign as the first term. Summing up all the terms with the coefficients $s_i$, and factoring out $R_x$ will yield the desired formula. Note that this same derivation applies to the left sided derivatives as well because the 'active' slopes of $T_{i+1}(W_i)$ are all the same whether a bend in $W_i$ is approached from the left or the right. The initial slope constant $L_x$ will just be different. $\qquad\square$

**Lemma A.3.** *If $E'_+(x) = E'_-(x)$, $E'(x)$ must be equal to 0.*

*Proof.* Let $S$ represent Equation 5, and $R$ and $L$ be the constants of proportionality for the directional derivatives. If $E'_+ = E'_-$, then $R_x S = L_x S$ for all $x \in P_i$. Since $W_i$ is comprised of alternating positive and negatively sloped line segments, $R_x$ and $L_x$ have opposite signs. The only way to satisfy the equation then is if $S = 0$. Consequently, $E'(x) = 0$ for all $x \in P_i$. $\qquad\square$

The following lemma shows that to calculate the derivative at of $F(x)$ for any bend point $x$, one needs only to compute the derivative of the finite approximation $f_i$ (which excludes $W_i$). This will be useful later for proving other results.

**Lemma A.4.** *For all $x \in P_i$:*

$$F'(x) = f'_i(x) = \sum_{j=0}^{i-1} s_j W'_j(x) \tag{6}$$

*Proof.* From the previous lemma we have $E'(x) = 0$ whenever the directional derivatives are equal. $F(x) = \sum_{j=0}^{i-1} s_j W_j(x) + E(x)$. The first $i - 1$ terms are differentiable at the points $P_i$ since those points lie between the discontinuities in $B_{i-1}$. Therefore $F'(x)$ is defined and can be calculated using the finite sum. A visualization of this lemma is provided in Figure 11. $\qquad\square$

We now prove our main theorem, which shows that there is a way to sum the triangular waveforms $W_i$ so that the resulting approximation converges to a continuously differentiable function. The idea of the proof is that much of the formula for $E'(x)$ will be shared between two successive generations of peaks. Once they are both valleys, they will be treated the same by the remaining compositions, so the sizes of their remaining discontinuities will need to be proportional.

**Theorem (3.1).** *$F'(x)$ is defined on $[0, 1]$ only if the scaling coefficients are selected based on $a_i$ according to:*

$$s_{i+1} = s_i(1 - a_{i+1})a_{i+2}$$

*Proof.* Rewriting Equation 5 (which is equal to 0) for layers $i$ and $i + 1$ in the following way:

$$s_i(1 - a_{i+1}) = s_{i+1} + \frac{1}{a_{i+2}}\left(s_{i+2} + \sum_{n=i+3}^{\infty} s_n \prod_{k=i+3}^{n} \frac{1}{a_k}\right)$$

$$s_{i+1}(1 - a_{i+2}) = s_{i+2} + \sum_{n=i+3}^{\infty} s_n \prod_{k=i+3}^{n} \frac{1}{a_k}$$

allows for a substitution to eliminate the infinite sum

$$s_i(1 - a_{i+1}) = s_{i+1} + \frac{1 - a_{i+2}}{a_{i+2}} s_{i+1}$$

Collecting all the terms gives

$$s_{i+1} = \frac{s_i(1 - a_{i+1})}{1 + \frac{1 - a_{i+2}}{a_{i+2}}}$$

which simplifies to the desired result. $\qquad\square$

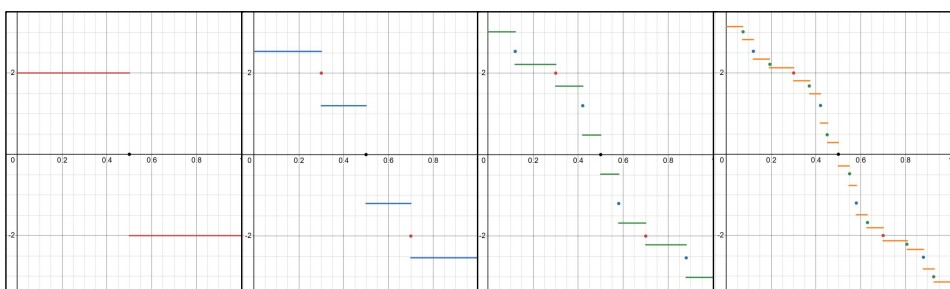

Figure 11: The derivatives of the first few stages of approximation. Notice that each time a constant segment "splits", the two neighboring segments adjacent to the split monotonically converge back to the original value (marked with a colored point corresponding to the step it's from)

### A.4    SUFFICIENCY FOR DIFFERENTIABILITY

This form of the scaling equation derived in the previous section is rather interesting. Since the ratio of two successive scaling terms is $(1 - a_{i+1})a_{i+2}$, factors of both $a_i$ and $1 - a_i$ are present in $s_i$. This has a few implications, firstly if any $a_i$ is 0 or 1, subsequent scales will all be 0, essentially freezing the corresponding neural network at a finite depth. Secondly, having both $a_i$ and $1 - a_i$ will cancel whichever slope multiplier $T_i$ contributes to $W_i$ at each point $x$, leaving behind the other term, which is less than 1.

If we ensure each $a_i$ is bounded away from 0 or 1 by being drawn from the an interval such that $c < a_i < 1 - c$ for some $0 < c < 0.5$, then the maximum value of $W_i'$ is always upper bounded by $(1 - c)^i$, which is sufficient for uniform convergence. This condition also means that the bends points (and thus the activation regions of the corresponding neural network) will become dense in $[0, 1]$, as each region is partitioned (at worst) in a $c : 1 - c$ ratio.

Lastly, we can show that bounds on $a_i$ and our choice of scaling values are sufficient for the existence of $F'$ on the bend points, in addition to being necessary for the existence of the derivative on the bend points, our choice of scaling is sufficient when $a_i$ are bounded away from 0 or 1.

**Theorem A.5.** *on bend points $x$, $F'(x)$ exists if we can find $c > 0$ such that $c \leq a_i \leq 1 - c$ for all $i$ and choose all $s_i$ according to Equation 3.1.*

*Proof.* We begin by considering Equation 5 for layer $i$ (it equals 0 by Theorem A.3).

$$s_i = \frac{1}{1 - a_{i+1}} \left( s_{i+1} + \sum_{n=i+2}^{\infty} s_n \prod_{k=i+2}^{n} \frac{1}{a_k} \right)$$

We will prove our result by substituting Equation 3.1 into this formula, and then verifying that the resulting equation is valid. First we would like to rewrite each occurrence of $s$ in terms of $s_i$. Equation 3.1 gives a recurrence relation. Converting it to an non-recursive representation we have:

$$s_n = s_i \left( \prod_{j=i+1}^{n} 1 - a_j \right) \left( \prod_{k=i+2}^{n+1} a_k \right) \tag{7}$$

When we substitute this into Equation 5, three things happen: each term is divisible by $s_i$ so $s_i$ cancels out, every factor in the product except the last cancels, and $1 - a_{i+1}$ cancels. This leaves

$$1 = a_{i+2} + (1 - a_{i+2})a_{i+3} + (1 - a_{i+2})(1 - a_{i+3})a_{i+4} + ... = \sum_{n=i+2}^{\infty} a_n \prod_{m=i+2}^{n-1} (1 - a_m) \tag{8}$$

This equation has a meaningful interpretation that is important to the argument. 1 is the full size of the initial derivative discontinuity at a point in $P_i$, and each term on the other side represents

proportionally how much the discontinuity is closed for each triangle wave that is added. Every time a wave is added, it subtracts the first term appearing on the right hand side. The following argument shows that each term of the sum on the right accounts for a fraction (equal to $a_i$) of the remaining discontinuity, guaranteeing its disappearance in the limit. Inductively we can show:

$$1 - \sum_{n=i+2}^{j} a_n \prod_{m=i+2}^{n-1} (1 - a_m) = \prod_{m=i+2}^{j} (1 - a_m) \qquad (9)$$

In words this means that as the first term appearing on the right in Equation 8 is repeatedly subtracted, that term is always equal to $a_n$ times the left side. As a base case, we have $(1 - a_{i+2}) = (1 - a_{i+2})$. Assuming the above equation holds for all previous values of $j$

$$1 - \sum_{n=i+2}^{j+1} a_n \prod_{m=i+2}^{n-1} (1 - a_m)) = 1 - \sum_{n=i+2}^{j} a_n \prod_{m=i+2}^{n-1} (1 - a_m)) - a_{j+1} \prod_{m=i+2}^{j} (1 - a_m)) =$$

using the inductive hypothesis to make the substitution

$$\prod_{m=i+2}^{j} (1 - a_m)) - a_{j+1} \prod_{m=i+2}^{j} (1 - a_m)) = \prod_{m=i+2}^{j+1} (1 - a_m))$$

Since all $c < a_i < 1 - c$, the size of the discontinuity at the points $P_i$ is upper bounded by the exponentially decaying series $(1 - c)^n$, which approaches zero. $\qquad \square$

### A.5 ERROR DECAY

**Lemma A.6.** *The ratio $s_{i+2}/s_i$ is at most $0.25$.*

*Proof.* by applying Equation 3.1 twice, we have

$$s_{i+2} = s_i(1 - a_{i+1})(1 - a_i + 2)a_{i+2}a_{i+3}$$

To maximize $s_{i+2}$ we choose $a_{i+1} = 0$ and $a_{i+3} = 1$. The quantity $a_{i+2} - a_{i+2}^2$ is a parabola with a maximum of $0.25$ at $a_{i+2} = 0.5$. $\qquad \square$

Since each $W_i$ takes values between 0 and 1, its contribution to $F$ is bounded by $s_i$. Since the $s_i$ decay exponentially, one could construct a geometric series to bound the error of the approximation and arrive at an exponential rate of decay.

### A.6 SECOND DERIVATIVES

Here we show that any function represented by one of these networks that is not $y = x^2$ does not have a continuous second derivative, as it will not be defined at the bend locations. To show this we will sample a discrete series of $\Delta y/\Delta x$ values from $F'(x)$ and show that the limits of these series on the right and left are not the same (unless all $a_i = 0.5$), which implies that $F''(x)$ does not exist (see Figure 12 below). First we will produce the series of $\Delta x$. Let $x$ be the location of a peak of $W_i$, and let $l_n$ and $r_n$ be its immediate neighbors in $B_{i+n}$.

**Lemma A.7.** *If $c < a_i < 1 - c$ for all $i$, we have $\lim_{n\to\infty} r_n = \lim_{n\to\infty} l_n = x$. Furthermore, $r_n, l_n \neq x$ for any finite $i$.*

*Proof.* Let $R$ and $L$ denote the magnitude of $W_i'$ on the left and right of $x$. $x$ is a peak location of $W_i$, so the right side slope is negative and the left is positive. Solving for the location of $T_{i+1}(W_i(x)) = 1$ on each side will give $l_1 = x - (1 - a_{i+1})/L$ and $r_1 = x + (1 - a_{i+1})/R$.

On each subsequent iteration $i + n$ ($n \geq 2$), $x$ is a valley point and the $\Delta x$ intervals get multiplied by $a_{i+n}$. Since $x$ is a valley point the right slope is positive and the left is negative. The slope magnitudes are given by $\frac{1}{x - l_n}$ and $\frac{1}{r_n - x}$ since $W_{i+n}$ ranges from 0 to 1 over these spans. Solving

for the new peaks again will give $l_{n+1} = x - a_{i+1}(x - l_n)$ and $r_{n+1} = x + a_{i+1}(r_n - x)$. The resulting non-recursive formulas are:

$$x - l_n = \frac{1 - a_{i+1}}{L} \prod_{m=2}^{n} a_{i+m} \text{ and } r_n - x = \frac{1 - a_{i+1}}{R} \prod_{m=2}^{n} a_{i+m} \tag{10}$$

The right hand sides will never be equal to zero with a finite number of terms since $a$ parameters are bounded away from 0 and 1 by $c$. $\qquad\square$

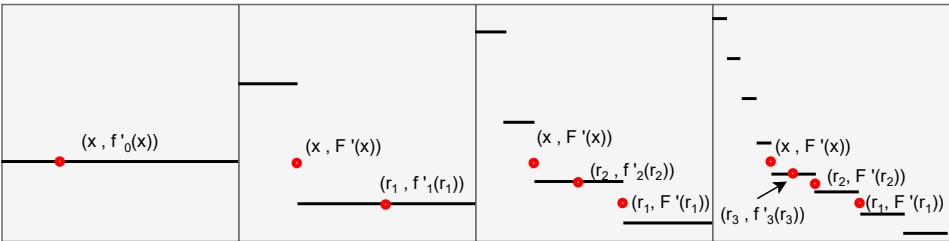

Figure 12: An illustration of attempting to calculate the second derivative. The points in the series approaching $x$ from the right are marked. We rely on the fact that at bend points, the first derivative converges back to the value it had at a finite point in the approximation. $a_0 \neq 0.5$ and all other parameters are set to $0.5$, which will cause the left and right sets of points to lie on lines with different slopes.

Next we derive the values of $\Delta y$ to complete the proof.

**Theorem A.8.** $F(x)$ *cannot be twice differentiable unless* $F(x) = x^2$.

*Proof.* The points $l_n$ and $r_n$ are all peak locations, Equation 6 gives their derivative values as $f'_{i+n}(r_n)$. In our argument for sufficiency, we reasoned about the sizes of the discontinuities in $f'$ at $x$. Since $l_n$ and $r_n$ always lie on the linear intervals surrounding $x$ as $n \to \infty$, we can get the value of $f'_i(x) - f'_{i+n}(r_n)$ using Equation 9 with the initial discontinuity size set to $Rs_i$ rather than 1. Focusing on the right hand side we get:

$$f'_i(x) - f'_{i+n}(r_n) = R * s_i \prod_{m=2}^{n} (1 - a_{i+m})$$

taking $\Delta y / \Delta x$ gives a series:

$$\frac{R^2 s_i}{(1 - a_{i+1})} \prod_{m=i+2}^{n} \frac{1 - a_m}{a_m}$$

The issue which arises is that the derivation on the left is identical, except for a replacement of $R^2$ by $L^2$. The only way for these formulas to agree then is for $R^2 = L^2$ which implies $a_i = 1 - a_i = 0.5$. Since this argument applies at any layer, then all $a$ parameters must be $0.5$ (which approximates $y = x^2$). $\qquad\square$

### A.7 MONOTONICITY AND CONTINUITY OF DERIVATIVES

Each of the $f'_i$ are composed of constant value segments, we will show that those values are monotonically decreasing (this can be seen in Figure 11). This can extend into the limit to show that $F'$ is monotone decreasing and that $F$ is concave.

**Lemma A.9.** *The function $F(x)$ is concave when all $s_i$ are chosen according to Equation 3.1.*

*Proof.* To establish this we will introduce the list $Y'_i = [F'(V_i[0]), f'_i(V_i[n]), F'(V_{i+1}[2^i])]$ for $0 \leq n \leq 2^i$, which tracks the values of $F'$ at the $i^{th}$ set of valley points. All but the first and last points will have been peaks at some point in their history, so Equation 6 gives the value of those

derivatives as $f_i'$.

We establish two inductive invariants. One is that the y-values in the list $Y_i$ remain sorted in descending order. The other is that $Y_i'[n] \geq f_i'(x) \geq Y_i'[n+1]$ for $V_i[n] < x < V_i[n+1]$, indicating that the constant value segments of $f_i$ lie in between the limits in the list $Y_i$. Together, these two facts imply that each iteration of the approximation $f_i$ is concave. Which then be used to prove that their limit $F$ is also concave.

As the base case $f_0$ is a line with derivative 0, $V_0$ contains its two endpoints. $Y_0'$ is positive for the left endpoint (negative for right) since on the far edges $F'$ is a sum of a series of positive (or negative) slopes, Therefore both the points in $Y'$ are in descending sorted order. The second part of the invariant is true since 0 is in between those values.

Consider an arbitrary interval $(V_i[n], V_i[n+1])$ of $f_i$, this entire interval is between two valley points, so $f_i'$ (which hasn't added $W_i$ yet) is some constant value, which we know from the second inductive hypothesis is in between $Y_i'[n]$ and $Y_i'[n+1]$. The point $x \in P_i \cap (V_i[n], V_i[n+1])$ will have $F'(x) = f_i'(x)$, and it will become a member of $V_{i+1}$. This means we will have $Y_{i+1}[2n] > Y_{i+1}[2n+1] > Y_{i+1}[2n+2]$, maintaining sorted order of $Y'$.

Adding $s_i W_i$ takes $f_i$ to $f_i + 1$ splitting each constant valued interval in two about the points $P_i$, increasing the left side, and decreasing the right side. Recalling from the derivation of Equation 5 all terms but the first in the sum have the same sign, so the limiting values in $Y_i'$ are approached monotonically. Using the first inductive hypothesis, we have on the left interval $Y_i'[n] = Y_{i+1}'[2n] > f_{i+1}' > f_i' = Y_{i+1}'[2n+1]$ and on the right we have $f_i' = Y_{i+1}'[2n+1] > f_{i+1} > Y_i'[n+1] = Y_i'[2n+2]$. And so each constant interval $f_{i+1}$ remains bounded by the limits in $Y_{i+1}'$.

We will now show by contradiction that the limit $F$ of the sequence of concave $f_i$ is also concave. Assume that $F$ is non-concave. Then there exist points $a$, $b$, and $c$ such that $F(b)$ lies strictly below the line connecting the points $(a, F(a))$ and $(c, F(c))$. Lets say it's below the line by an amount $\epsilon$. Since at each point $f_i$ converges to $F$, we can find $i_a$ such that $f_{ia}(a) - F(a) < \epsilon/2$, etc..., we take $i = \max(i_a, i_b, i_c)$. Since $f_i(a)$ and $f_i(c)$ are no more than $\epsilon/2$ lower than their limiting values, the entire line connecting $(a, f_i(a))$ and $(c, f_i(c))$ is no more than $\epsilon/2$ lower than the line between $(a, F(a))$ and $(c, F(c))$. $f_i(b)$ is also no more than $\epsilon/2$ higher than $F(b)$, thus $f_i(b)$ must still lie below the line between $(a, f_i(a))$ and $(c, f_i(c))$, making $f_i$ non-concave and producing a contradiction. $\qquad\square$

Lastly we briefly sketch out why $F'$ is continuous. It relies on some of our earlier reasoning. Monotonicity of the derivative makes continuity easy to show, because when $x_1$ within $\delta$ of $x_2$ has $f(x_1)$ within $\epsilon$ of $f(x_2)$, so do all intermediate values of $x$. We can establish continuity of $F'$ at bend points $x$ easily by using Equation 9. We can pick an $i$ so that the constant value segments of $f_i'$ are within $\epsilon$ of $F'(x)$ and then use the next iteration of bend points (since the constant intervals split, but the new segments near $x_n$ converge monotonically towards it) to find $\delta$. In the case of continuity for non bend points $x$, they sit inside a constant-valued interval of $f_i$ for each $i$. We can choose $i$ such that $\sum_{n=i}^{\infty}(1-c)^n < \epsilon/2$ because this series constrains how far derivative values can move in the limit, and then use the constant interval $x$ is situated in to find $\delta$.

## A.8 LEARNING RATES

All results in the main body of the paper used a constant learning rate of $10^{-3}$. In this appendix, we considered an ablation study on the learning rate for the task of learning $y = x^3$. As seen in Figure 13, the learning rate we selected was approximately optimal for both our method as well as default network training. We note that for this ablation study, a constant 1000 epochs were run, which explains why both methods perform worse as the learning rate becomes minuscule. At small learning rates, what is really measured is how close of a guess the initialization is to the target function. Our networks are preforming better here simply because they are always outputting convex

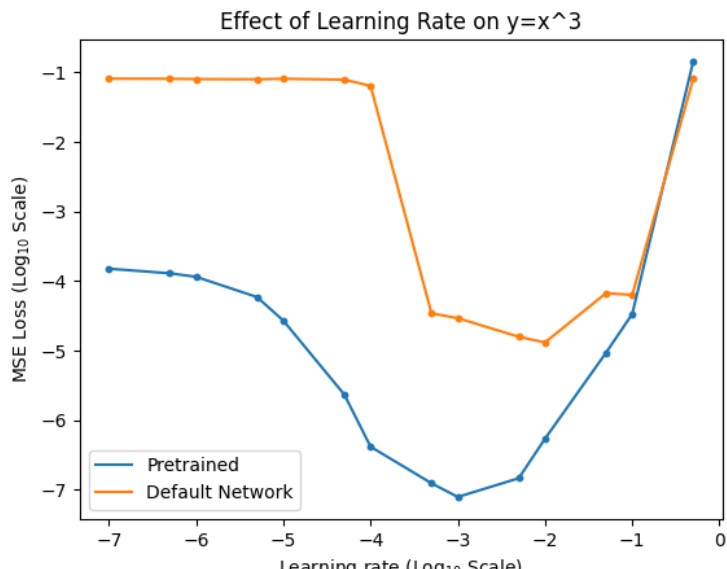

Figure 13: Learning $y = x^3$ at various learning rates for 1000 epochs. The figure shows the log losses of the best model of 30 for networks trained according to the methods in this paper versus random initialization and gradient descent ("Default Network"). For high learning rates, neither learns because the steps are too large. For low learning rates (below $10^{-5}$), the steps are too small, and our networks are likely deriving an implicit advantage from being forcibly initialized to a convex function. Both methods are able to converge for learning rates in between $10^{-4}$ and $10^{-2}$; one could run for more epochs to see a similar advantage of our method for smaller learning rates, if desired.

functions. But this only accounts for losses on the order of $10^{-4}$, which indicates that in the more reasonable learning rate ranges, our pretraining is performing a meaningful function and enabling order-of-magnitude improvements over default network training.

## A.9 REAL-WORLD DATA AND CLASSIFICATION PROBLEMS

Here we present a few preliminary results on extending our pretraining technique to classification problems and real-world datasets. The classification problem we chose is the classic two spirals dataset, and we selected the UCI dataset "Concrete Compressive Strength" (Yeh, 1998) for our real-world regression task. The concrete dataset has 8 numerical features that can be used to predict the compressive strength of a concrete sample.

The networks are set up as described before, with our 4-neuron-wide networks acting as 1D to 1D activation functions inside randomly initialized standard linear layers. In these experiments, we use one "hidden layer" of our networks (which we choose to have depth 5 in our tests). The geometric interpretation of this is that we are setting up one-dimensional convex functions oriented in random directions on the input space, and then taking a linear combination of them as the network output. In the case of classification, the loss function is simply swapped for cross entropy.

After our pretraining phase, there are two choices of how to conduct the second phase of training (training matrix entries directly). All the parameters could be freed from constraints ("dense"), or the smaller 4-neuron subnetworks could be kept isolated from each other (but otherwise have their parameters freed). In the case where the 4-neuron networks remain in isolation, the weight matrices of the hidden layers will have a block diagonal structure (block size 4—the width of each subnetwork). Thus, we consider two fair (same number of free parameters) comparisons in this subsection: (1) dense versions of our and Kaiming-initialized networks, and (2) block diagonal versions of our and Kaiming-initialized networks. As discussed below and in the figure captions, we find that while the dense variant of our method ties or is slightly worse than a regular fully-connected network of the same dimensions, in the block diagonal case, our experimental networks significantly

outperform their Kaiming-initialized counterparts. This makes sense in light of the experiments from the main body, where we can effectively shape the output of 4-neuron-wide networks better than random initializations, but where ensembles of our networks (see approximation of $x^3 - x$ in Figure 7) get filled in with noise by gradient descent during the second training stage. This again highlights the need for more mathematical developments to provide a better extension into higher-dimensional nonconvex functions.

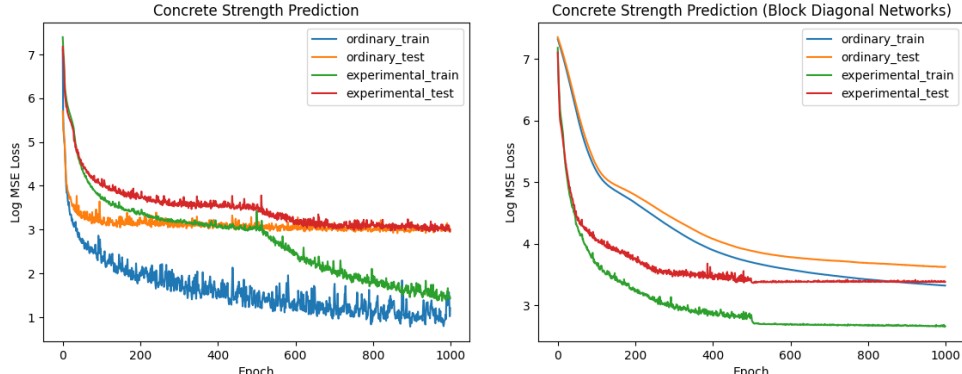

Figure 14: Train/test loss plots for Kaiming-initialized ("ordinary") fully-connected networks and our method ("experimental"). Our method has two steps: pretraining on triangle peaks, followed by the optimization of the raw matrix weights. The switch-off is at epoch 500, hence the associated visible change in the loss curves. The block diagonal variants of the networks (right) are generally worse at the task (test losses 30.1 (ours) and 40.8 (Kaiming)) than the dense variants (losses of 22.3 (ours) and 21.2 (Kaiming)). Our experimental networks outperform by up to an order of magnitude in the block diagonal case. We used 32 of our convex blocks (i.e., all weight matrices are size 128, including for standard comparison networks).

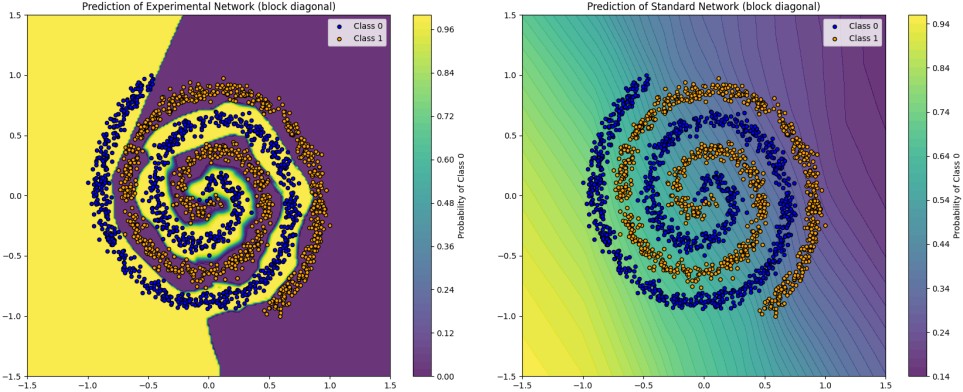

Figure 15: Class predictions of block diagonal networks on a standard two-spiral dataset; ours (left) and Kaiming-initialized (right). While the cross-entropy losses are comparable for the dense-matrix variants of both networks, when a block diagonal structure is imposed, Kaiming initialization fails to learn the spiral. Note that the colors of the network class predictions are inverted for visibility. Cross-entropy losses are 0.0032 and 0.63 respectively. We used 16 of our convex blocks, so all weight matrices are 64 by 64.

Nonetheless, while we believe there is much room for future work to improve higher-dimensional results, Figures 14 and 15 already show impressive results using our current approach. In the case of the concrete problem (Figure 14), our experimental network outperforms a standard Kaiming-initialized network when block diagonality is enforced. In the case of the two-spiral classification task (Figure 15), our network is able to learn an accurate decision boundary (left subfigure), whereas

a standard network constrained to be block diagonal fails to learn (right subfigure). When dense weight matrices are trained, our networks are almost able to tie with the standard initializations, although convergence can take longer. These results suggest that our method might be quite powerful when its parameter budget is spent on greater width and depth of convex function blocks (as in Figure 9) instead of filling in dense matrices.

Our two-step training approach adds some practical challenges, such as deciding when to switch parameterizations for optimal convergence time. Additionally, since the optimization variables are different, the optimizer will have to restart any momentum or adaptive learning rates when the switch is made, which can sometimes cause loss to temporarily spike. We lowered the learning rate on the second step to avoid this. Further interesting optimizations of our approach (e.g., training another network to inform our algorithm when to switch parameterizations) are imagined as future work.

## POTENTIAL BROADER IMPACT

This paper presents work whose goal is to enable more efficient neural networks. While the present work is largely theoretical, future advances in this line of research could enable the use of much smaller networks in many practical applications, which could substantially mitigate the rapidly growing issue of energy usage in large learning systems.

