# OpenReview forum: "Compelling ReLU Networks to Exhibit Exponentially Many Linear Regions at Initialization and During Training"
_ICLR.cc/2025/Conference — Submitted to ICLR 2025_

### Official Review · Reviewer_rFNb · 2024-10-27

**Soundness:** 3
**Presentation:** 2
**Contribution:** 2
**Rating:** 5
**Confidence:** 3

**Summary:**

This paper presents a novel strategy to improve the efficiency of ReLU neural networks. It focuses on overcoming the limitations of randomly initialized networks, which tend to be unnecessarily large and inefficient in approximating simple functions. The authors introduce a reparameterization of network weights that ensures an exponential number of activation patterns, thus maximizing the linear regions in the input space. Their approach includes a pretraining stage using derived parameters that enhances the expressivity of the network before standard gradient descent is applied. This method shows significant improvement in approximating both convex and non-convex functions, with better accuracy and efficiency compared to traditional networks. The paper's findings demonstrate that networks initialized with exponential linear regions can capture nonlinearity more effectively, leading to more accurate function approximations. It concludes with potential extensions to multidimensional and non-convex functions, positioning this strategy as a promising tool for more efficient deep learning models.

**Strengths:**

* The paper introduces a novel approach to reparameterize ReLU network weights, which forces the network to exhibit an exponential number of activation regions. This significantly enhances the expressivity of the network and addresses the inefficiencies of randomly initialized models, providing a more accurate and efficient approximation of nonlinear functions.

* The proposed pretraining strategy allows the network to initialize with exponentially more linear regions, thus reducing the reliance on gradient descent to discover new activation patterns. This results in faster convergence and much more accurate function approximations, as demonstrated through numerical experiments, showing orders of magnitude lower errors compared to standard initialization methods.

**Weaknesses:**

* While the paper demonstrates improvements in one-dimensional convex functions, the results for higher-dimensional functions and complex non-convex problems are not as thoroughly explored. The proposed method may face scalability challenges when extending to high-dimensional inputs, where the complexity of real-world tasks lies.

* The introduction of a pretraining step with specific reparameterization adds complexity to the network training pipeline. This may make the approach more difficult to implement or integrate into standard deep learning workflows, especially for practitioners looking for more straightforward techniques.

* The effectiveness of the method heavily relies on carefully derived theoretical constructs, such as triangle functions and their parameterization. While this works well in controlled scenarios, its practical robustness in more diverse and noisy real-world datasets is not fully tested or demonstrated.

**Questions:**

* How does the proposed reparameterization strategy perform in complex, high-dimensional tasks, and what are the challenges in scaling this method effectively to more realistic datasets?

* Have you considered testing this method on real-world datasets with more variability and noise? How robust is the technique in such scenarios, and are there any performance trade-offs when dealing with non-synthetic data?

---

> ### Author Response · Authors · 2024-11-22
>
> We respectfully decline to publicly respond to this review and have instead sent private comments to the Area Chairs. We await their reply before taking any further action.

---

> ### Author Response · Authors · 2024-11-26
>
> Following the advice of our AC, and as a professional courtesy, we will avoid publicly mentioning the obvious issues with the above review text, and we will attempt to respond to the points in the text as-is.
>
> W1: The point of our paper is an innovative, constructive approach to neural network initialization and training.  We are up front in the paper that our examples are primarily low-dimensional, yet we argue that any paper that makes any theoretically-proven progress towards *exponentially* more efficient neural networks (as our work does) is worthy of being shared with the ICLR community.  Nonetheless, per the suggestions of Reviewers oevS and hKZ8, we have added in our rebuttal more “real-world” examples, including separating two spirals (classification) and an 8-D regression problem using a standard real-world dataset from a UCI repository.
>
> W2: The pseudocode for our algorithm is provided in our manuscript, and despite the sophisticated mathematics used to arrive at our method, the resulting pseudocode is very simple.  Just like any other initialization strategy (Kaiming, RAAI, etc.), we would release our code as open-source, and other people could simply use it.  Our method adds no more complexity than any other initialization strategy, and in the reported tests, we can produce far better results.
>
> W3: See response to W1.  We have added real-world examples including noisy real-world datasets (the UCI concrete dataset example, and a noisy two-spiral dataset).
>
> Q1: Challenges in scaling the method are already discussed in the last section of the paper, and now there are mentions in new sections like Appendix A.9 as well.
>
> Q2: See above regarding new examples.
>
> We are confident that our revised manuscript has overcome any potential issues mentioned in the above text.  We feel that each numerical rating provided in the above text is inaccurate (e.g., all our theoretical results are proven and then numerically verified; they are as sound as possible).  Again, we will refrain from further public commentary on this review text but are happy to respond to any comments the reviewer may wish to write below.  And nonetheless, in the spirit of the rebuttal period, we would be grateful if the reviewer reconsidered their scores in light of our substantially revised manuscript.

---

> > ### Comment · Reviewer_rFNb · 2024-12-01
> > **RE:**
> >
> > I'd like to thank the authors for addressing some of my concerns in the response. I have increased my score based on the authors' responses.

---

> > > ### Author Response · Authors · 2024-12-02
> > >
> > > We appreciate the reviewer’s increase in their score after our first round of significant improvements.  We would like to direct the reviewer to our second round of major improvements, which include evaluations of our method on CIFAR-10 and ImageNet.  Please see the overall comment we’ve added.
> > >
> > > We re-read the comments posted by the reviewer and are not sure of any outstanding criticisms or questions we have not addressed.  Beyond this, we have now added far more complex examples (CIFAR-10 and ImageNet) to our paper, addressing the suggestions of reviewer EGry; and our method is able to outperform Kaiming initialization on these benchmarks.  In light of this, we kindly request the reviewer revisit their score, or post any questions or criticisms that are causing the reviewer to be on the fence.  Suggestions for improvement are welcomed and will be incorporated as feasible in the remaining discussion period.

---

### Official Review · Reviewer_hKZ8 · 2024-10-31

**Soundness:** 3
**Presentation:** 3
**Contribution:** 3
**Rating:** 8
**Confidence:** 3

**Summary:**

This paper proposes to reparameterize ReLU networks by parameterizing the peaks of the triangle wave basis functions generated by ReLU activations. This ensures that the number of linear regions grows exponentially with the depth of the network, which reduces the waste of representation capacities in randomly initialized ReLU networks. A learning algorithm is proposed to train the reparameterized ReLU network by first updating the derived parameters and then updating the actual weights underlying the model. The proposed method is empirically evaluated on 1D convex and 2D nonconvex target functions, which demonstrate its improved accuracy compared to randomly initialized networks.

**Strengths:**

1. The proposed reparameterization of ReLU networks is novel and interesting. Based on the observation that ReLU networks can generate symmetric triangle waves, the proposed approach introduces an approach to directly reprent the ReLU neurons (rather than weights) as asymmetric triangle wave basis function with learnable locations of the peaks within [0, 1].
2. The proposed learning algorithm is simple and seems to be effective in training the reparameterized network for simple target functions as shown in the experiments.
3. The 1D demonstrations and experiments are nice, which illustrate how the proposed method learns useful patterns for fitting the target function.

**Weaknesses:**

1. Theorem 3 seems to be an “only if” statement, and therefore setting $s_{i+1}$ according to Eq 4 is a necessary but not sufficient condition to guarantee differentiability of the reparameterized network.

2. It is unclear how useful the proposed method is in practice, since it is mostly evaluated on simple 1D convex target functions. As this is not a theory paper, the proposed method should at least be evaluated on some semi-real datasets, (e.g., some of the common UCI datasets).

3. Writing needs to be improved. The logic flow is a bit confusing. Also, several important concepts and building blocks are not well explained in the paper. For example, important definitions like definitions of linear regions, activation patterns, and activation regions should be stated in the paper or appendix.

4. The following closely related works which analyze compositions and/or reparameterization of ReLU activations are not discussed in the paper.

[1] K Eckle, J Schmidt-Hieber. A comparison of deep networks with relu activation function and linear spline-type methods. Neural Networks 2019.

[2] DM Elbrächter, J Berner, P Grohs. How degenerate is the parametrization of neural networks with the ReLU activation function? NeurIPS 2019.

[3] W Chen, H Ge. Neural characteristic activation analysis and geometric parameterization for ReLU networks. NeurIPS 2024.

[4] B Hanin, D Rolnick. Complexity of linear regions in deep networks. ICML 2019.

[5] M Raghu, B Poole, J Kleinberg, S Ganguli, J Sohl-Dickstein. On the expressive power of deep neural networks. ICML 2017.

[6] D Rolnick, K Kording. Reverse-engineering deep ReLU networks. ICML 2020.

**Questions:**

1. How are ReLU activations guaranteed to generate symmetric triangle waves? There are many possible compositions of ReLU activations, but only a subset of them are symmetric triangle waves. If the network is reparameterized using only the triangle wave basis functions as proposed in the paper, will it lose some flexibility and expressivity as it is not possible for the reparameterized network to create other shapes or patterns within each layer?

2. Could you provide some intuitions and/or theory regarding why pretraining helps maintain the triangle generating structure and avoid eliminating activation regions as the network gets deeper?

3. Does the proposed method improve convergence rate? Could you demonstrate it with some experiments and/or theory?

---

> ### Author Response · Authors · 2024-11-22
>
> W1: You’re correct that this is an ‘only if’ statement. The sum coefficients on the triangle waves need to follow the a1*(1-a2) pattern if there is to be any chance of getting a differentiable-everywhere sum in the limit. But there are still ways you can mess it up: for example if you choose a peak location to be 0 or 1, you’ll be stuck with a piecewise linear sum in the limit. In the appendix (where most of the math is derived, and will be improved in our revision) we talk about how selecting the peak locations in an interval like (0.1, 0.9) is enough to complete the ‘if’ part of the statement. This stops the collapse at 0 or 1, and also prevents awkward sequences like 1 / 2, 2 / 3, 3 / 4, 4 / 5, 5 / 6, etc… where one linear region is only decaying in size at a rate of 1/n instead of exponential. Again, we will clarify this in our revision, and we appreciate your careful reading of this theorem.
>
> W2: We feel obligated to say that this is a theory paper. It derives a way to get exponential use out of depth, while preventing the network from outputting a fractal, and it translates this into a training algorithm. The value of this paper is that it demonstrates that by taking a more principled approach to neural networks, it’s possible to realize up to an exponential gain in efficiency. Our experiments are kept simple for this purpose, they serve only to complement the theory. We hope the reviewer may appreciate this perspective and reconsider the paper in light of this view.
>
> W3: Although we cite a paper on the first page of our manuscript that defines the terms the reviewer mentions, we are also happy to add such definitions to our appendix.
>
> Q1: The reparameterization we use does restrict the flexibility of the networks, as only certain combinations of weights can make triangle waves. But much of the flexibility we eliminate is the flexibility to be inefficient. Furthermore we show that the triangle waves can be of some use when it comes to approximating differentiable convex functions. The importance of doing this is that neural networks will not find the weights to make the triangle waves (or any efficient shape) on their own (see the hanin and rolnick paper cited in the introduction) so we essentially do it for them.
>
> Q2: The pretraining is useful because it moves the approximation produced by the network closer to the target function before turning it over to (naive) gradient descent. Since the loss function carries no direct information about the number of linear regions the network has, it does not encourage the network to learn an efficient setting of weights, just one that gets close to the target function. The elegant thing about the pretraining and the triangle parameterization is that it ensures an exponential number of linear regions the whole time, and if it can get close enough to the target function, those regions will still be useful once the pretraining constraints are lifted.
>
> Q3: We could certainly look at the convergence rates, thank you for the suggestion. The reason we haven’t focused on it so far is that the goal of this reparameterization is to make the network find more efficient and accurate solutions. Whereas other reparameterizations, such as the third paper you suggested citing (the geometric parameterization one) are more focused on improving the training dynamics. On the subject of the GmP paper, our pretraining stage is likely immune to the instability issues that standard parameterization and weight norm display since we do not use any form of weight decay, so small perturbations will never be on the same magnitude as our weights. Nor do we need to use any form of normalization like batch or layer norm, the math naturally led our weight setting scheme into a resnet-like structure (the sum acts as half a residual, it keeps a sum, but does not connect back to the triangle waves, as giving them a monotone function would hinder their linear region production) that does its own normalization. The scaling factors needed for differentiable functions to be produced are naturally exponentially decaying.
>
> One last observation about GmP paper is that the conversion from hyperspherical coordinates needs to be done serially to be done in linear time. Also, we did not see an analysis of the complexity of the backpropagation; while it is possible to backpropagate and update the hyperspherical parameters without adding additional big-O costs, doing so is again a serial operation (and very complicated). We also don’t elaborate much on the specifics of the backpropagation in this paper (although perhaps we ought to), which requires going backwards through how the weights were set based on triangle peaks (which involves some squaring and inversing), but in our case there is no serialization introduced, and each weight only depends on one or two different triangle peak parameters, so there is no danger of increased computational complexity.

---

> ### Comment · Reviewer_hKZ8 · 2024-11-23
>
> Thank you for your responses which clarify most of my concerns and questions.
>
> Regarding experiments, I do appreciate the authors' perspective, but I'd still be interested to see some results on some UCI datasets if that's possible, just to get a sense of how the proposed method performs on common benchmarks. It's completely fine if the results were not that great. But if that's the case, I'd appreciate some discussions on the possible discrepancies between the theory and practice, since it would be valuable to know the potential limitations of the proposed method when it comes to practical applications.
>
> Please add the discussions and clarifications in your rebuttal to the revision (especially those regarding W1, Q1, Q2 and GmP), and I will revise my rating accordingly.

---

> ### Author Response · Authors · 2024-11-26
>
> Dear reviewer hKZ8: Thank you again for your kind shepherding of our manuscript.  We wanted to quickly note that we've uploaded a revised PDF that addresses all of the feedback you provided.  We have (1) corrected unclear language around our math, (2) added meaningful comments on all the helpful citations provided (see related work, conclusion, and appendix), (3) added definitions (in the appendix) of any basic terminology we assumed, (4) added numerical experiments on noisy and higher-dimensional classification and real-world regression tests (thank you again for pointing us to UCI!) (Appendix A.9), and (5) plotted loss curves in Appendix A.9 that give a sense of how our networks converge compared to Kaiming initialization.  With these major improvements, along with those suggested by reviewer oevS, we are thankful for any reconsideration you may offer.  Furthermore, we would be grateful for any further suggestions from the reviewer that we can address during the author-reviewer discussion period, in order to ensure we are giving the best presentation of our work to the reviewers (and hopefully the community).  Thank you again.

---

> ### Comment · Reviewer_hKZ8 · 2024-11-26
>
> Thank you for your effort in addressing my comments. The revised version of the paper has addressed all my major concerns, and has improved a lot in terms of clarity, soundness and completeness. I do find the perspective of this work interesting and insightful, which is potentially useful for the community. Therefore, I have increased my ratings to recommend acceptance of this work.

---

### Official Review · Reviewer_oevS · 2024-10-31

**Soundness:** 3
**Presentation:** 3
**Contribution:** 3
**Rating:** 8
**Confidence:** 2

**Summary:**

This paper focuses on the expressivity of ReLU networks and argues that the standard neural network training approaches lead to models that cannot utilize all of the linear regions that a ReLU network has the potential to exhibit. The paper contains an approach to overcome this issue.

**Strengths:**

Please see the Questions section.

**Weaknesses:**

Please see the Questions section.

**Questions:**

My review is as follows:

- I think this paper brings up important issues with standard neural network training practices (such as using relu for activation or gradient descent, etc).

- One thing that I think is potentially missing is the verification of the findings on a somewhat more realistic scenario. Could we expect the proposed method to outperform a standard neural network approach (e.g. a similar size relu network trained by SGD) when, say, predicting airline delays? Or, other more standard methods such as linear regression or decision tree?

- To make the results potentially more broad, I wonder if the proposed strategy could be somehow applied to classification (perhaps can test it on a simple dataset such as "two spirals" dataset). If that's not straightforward, I think it'd still help me understand the contributions better if the authors can comment on the challenges.

- Which of the findings of this paper could we expect to carry over to other non-linear activation functions such as sigmoid?

---

> ### Author Response · Authors · 2024-11-22
>
> We appreciate the reviewer’s encouraging view of our work and their constructive feedback.  We address the reviewer’s feedback as follows:
>
> Q2: Our theoretical guarantees are for convex functions and non-convex functions that are differences of convex functions. While our method may perform well for functions outside those classes, such performance may not be theoretically guaranteed (and the primary emphasis of our method is to develop the mathematical theory of this training procedure, and simply demonstrate it with a few numerical test cases). That being said, convex functions arise often in real life - for instance, see Stephen Boyd’s famous, large textbook on convex optimization. So we believe that our demonstrations on learning convex (and non-convex) functions already gives a decent impression that our method can be used to learn realistic functions. If there is a particular regression function/dataset/problem the reviewer would like us to try, we’re more than happy to run the experiment and add the results to the revised manuscript.
>
> Q3: It’s possible to try a classification problem by assembling the width-4 networks into a larger network similar to how we approximate z = r^3 and y=x^3 - x, and then putting a cross entropy loss layer at the end or something. The reason we’re generally hesitant to try new experiments is that much more work needs to be put in on the math side to derive a more expressive pretraining parameterization. The primary limitation of our current method is that the pretraining doesn’t directly converge to any of the 1D functions we use it to approximate, as they lack a sufficient degree of symmetry, so we can’t show exponential performance gains out to arbitrary depth (it would still beat kaiming initialization, since at infinite depth the dying ReLU problem is guaranteed to happen). In summary: it’s possible to try classification problems, but the theoretical performance guarantees we’re able to prove with our small networks will likely not hold up for larger networks without additional research on extending those guarantees to networks of larger depth. We are willing to add such an experiment if it would raise the reviewer’s score, but we also hope the reviewer can understand our concerns; we are happy to discuss with the reviewer about this point in this thread.
>
> Q4: If another activation is used, ‘linear regions’ isn’t a well-defined measure of approximation power, as the approximation the network produces will no longer be piecewise linear. But there may be other ways to measure complexity (I’ve seen a few things that use topology, etc) that apply to nonlinear activations. The idea of creating oscillating hidden representations that fold the input domain on itself to maximize the use of the activation function’s nonlinearity is likely still relevant. The network parameterization presented in this paper likely can be reformulated in terms of leaky ReLU; the exact weights would just be different based on what the negative slope is. Thus, while it is possible many of our results could apply to non-linear activation functions, this would be a significant (albeit interesting!) amount of work for the future.
>
> We would be happy to discuss the paper further with the reviewer and address any other feedback that may influence the reviewer’s score.

---

> ### Author Response · Authors · 2024-11-26
>
> Dear reviewer oevS: We just wanted to note that we've submitted a revised version of the manuscript that incorporates the changes we proposed to address your questions and suggestions.  Most importantly, Q2 and Q3 are now answered with noisy, real-world, higher-dimensional experiments in our substantially revised paper (see Appendix A.9, which we've ensured is referenced from the main body).  We believe that these improvements, along with those thanks to reviewer hKZ8, have resulted in a significantly strengthened manuscript.  We are grateful for any amount this may influence the reviewer's score, and if there are further improvements the reviewer can imagine, we would be grateful to incorporate those in the remainder of the author-reviewer discussion period.

---

> > ### Comment · Reviewer_oevS · 2024-11-26
> >
> > Appendix A.9 looks good to me. Thanks for updating the manuscript. I'm happy to increase my score.

---

### Official Review · Reviewer_EGry · 2024-11-08

**Soundness:** 2
**Presentation:** 3
**Contribution:** 2
**Rating:** 3
**Confidence:** 4

**Summary:**

This paper designs a novel training strategy: (1) reparameterise the network weights in to make it exhibit a number of linear regions exponential in depth; (2) train on the derived parameters for an initial solution; (3) refine the parameter by directly updating the underlying model weights. Experiments are given to support the method.

**Strengths:**

-	The paper presents detailed introduction and explanation of the proposed method, including how to construct the initialisation and how to calculate the gradient. The paper is compressive, well-structed, and easy to follow.
-	The paper present experiments for cases of one dimension and high-dimension non-convex problems.
-	I find the paper makes conceptual contributions of proposing a new initialisation strategy.

**Weaknesses:**

-	The experiments are not sufficient. The current experiments only cover quite shallow (three layers) ReLU neural networks on very simple tasks. It is unclear whether the results apply to complex scenarios, like deeper neural networks, transformer on fitting images, mining on text data, etc. Thus, the paper actually cannot help understand the success of deep learning.
- No comparison is given with other initialisation methods.
- The explanation of why this method works is not sufficient. This makes the method not convincing.
-	No theoretical results are provided. This is particularly severe given the experiments are insufficient.

The paper looks like in an early stage with insufficient validation. I suggest the authors to do more following on this paper.

**Questions:**

Please address the Weaknesses.

---

> ### Author Response · Authors · 2024-11-22
>
> We respectfully decline to publicly respond to this review and have instead sent private comments to the Area Chairs. We await their reply before taking any further action.

---

> > ### Comment · Reviewer_EGry · 2024-11-25
> >
> > Thanks the authors for letting me know. Do they now want to take any further action? E.g., open discussion to defend their merit, if they value open peer review?

---

> ### Author Response · Authors · 2024-11-26
>
> We discussed this review privately with our AC, and per their advice, we will publicly respond to the provided points as-is.  As a professional courtesy, we will address these points with severe restraint:
>
> W1: “the paper actually cannot help understand the success of deep learning” The paper mathematically proves a new algorithm for initializing and pretraining ReLU networks so as to achieve *exponentially* better efficiency.  Not only does our paper help illuminate the mathematical foundations of deep learning, it advances them.  Furthermore, in our revision, we have added examples of separating two noisy spirals (classification) and doing a real-world, noisy, 8-D regression problem with a standard UCI dataset.  Our method is not related to text mining.
>
> W2: “No comparison is given” Comparisons are indeed given with other initialization methods - Kaiming and RAAI.  These are the first two rows, respectively, of Tables 1 and 2, for instance (see also Figure 4, Table 3, and Figure 5 vs. Figure 3).  Reflecting on this list, we realize that in fact, at least half of the results and figures in the main paper body illustrate comparisons with competing methods.  (And our comparisons often show our method outperforming by multiple orders of magnitude over the available methods in PyTorch.)  Additionally, the new examples in the revised manuscript provide even more comparisons against Kaiming-initialized networks.
>
> W3: “The explanation of why this method works is not sufficient. This makes the method not convincing.” One does not need to be convinced, as we are using rigorous mathematics, not rhetorical techniques.  The theoretical results that explain why our method works are all proven, and the reviewer may read the appendix if they would like to see the proofs.  If the reviewer finds that we are missing a theorem or proof that justifies the performance of our method, or if they find a flaw with one of our proofs, we would welcome such discussion.
>
> W4: “No theoretical results are provided.”  This is a theory paper with theoretical results throughout the body and appendix.
>
> We are disheartened by the reviewer's insinuation in a public forum that we do not value open peer-review, which is a personal remark unrelated to the merit of the present work.  We invite the reviewer to read our revised manuscript end-to-end and to make an authentic assessment of our work, including having the grace and honesty to correct their Confidence score.  We are willing to reply to any questions about the paper the reviewer may have.

---

> > ### Comment · Reviewer_EGry · 2024-11-27
> >
> > I welcome your response. Below is my inline reply:
> >
> > > The paper mathematically proves a new algorithm for initializing and pretraining ReLU networks so as to achieve exponentially better efficiency. Not only does our paper help illuminate the mathematical foundations of deep learning, it advances them.
> >
> > I have different expectations for a new algorithm that is claimed to be "mathematically proved".
> >
> > The presented "theory" is about calculating derivatives, derivatives' range, differentiability - none of them supports the algorithm's performance, or why and when the proposal algorithm is guaranteed and/or better than existing methods. If the paper is treated as a theory paper, I would be curious about the convergence, generalizability, and/or algorithmic stability of your algorithm.
> >
> > I also don't think the presented theory helps "illuminate the mathematical foundations of deep learning", because the theorems are only for the proposed method. Proving its differentiability doesn't help us understand deep learning.
> >
> > > Furthermore, in our revision, we have added examples of separating two noisy spirals (classification) and doing a real-world, noisy, 8-D regression problem with a standard UCI dataset. Our method is not related to text mining.
> >
> > I don't think experiments using a UCI dataset are enough for quantitatively validating a new method in deep learning. Please consider at least CIFAR-10/100 and ImageNet.
> >
> > > W2: “No comparison is given” Comparisons are indeed given with other initialization methods - Kaiming and RAAI. These are the first two rows, respectively, of Tables 1 and 2, for instance (see also Figure 4, Table 3, and Figure 5 vs. Figure 3). Reflecting on this list, we realize that in fact, at least half of the results and figures in the main paper body illustrate comparisons with competing methods. (And our comparisons often show our method outperforming by multiple orders of magnitude over the available methods in PyTorch.) Additionally, the new examples in the revised manuscript provide even more comparisons against Kaiming-initialized networks.
> >
> > Thanks for your clarification. This concern is partially cleared. If you hope to meaningfully compare your method with Kaiming and RAAI, please present comparisons in larger datasets for wider tasks.
> >
> > > W3: “The explanation of why this method works is not sufficient. This makes the method not convincing.” One does not need to be convinced, as we are using rigorous mathematics, not rhetorical techniques. The theoretical results that explain why our method works are all proven, and the reviewer may read the appendix if they would like to see the proofs. If the reviewer finds that we are missing a theorem or proof that justifies the performance of our method, or if they find a flaw with one of our proofs, we would welcome such discussion.
> >
> > Explanation is indeed required, especially given the proposed algorithm's performance is not guaranteed either theoretically or empirically.
> >
> > Re “using rigorous mathematics”- please see my feedback above. Your proofs may be “flawless” but do not show your algorithm has advantages.
> >
> > > W4: “No theoretical results are provided.” This is a theory paper with theoretical results throughout the body and appendix.
> >
> > Thanks for your clarification. Please let me redress my comments: no theoretical results on the performance are provided. Proving differentiability (in the main text) is far away from sufficiency for a learning theory paper.
> >
> > If the authors think this is a theory paper, please present convergence, generalisation, and/or algorithmic stability properties of your algorithms; please show your algorithm ‘s performance is theoretically guaranteed and advanced.

---

> > > ### Author Response · Authors · 2024-11-29
> > >
> > > We first would like to thank the reviewer for their thoughtful comments here.  We appreciate the time it takes to review a manuscript, and we value all constructive feedback to identify the limitations of (and ultimately strengthen) our manuscript.  We’re very happy to discuss the new and restated points the reviewer has provided:
> > >
> > > 1. We first mention one potential terminology issue that we think may have caused mismatched expectations.  Our paper theoretically develops (and numerically validates) a mathematically principled algorithm.  Our focus is on the mathematical ideas that lead to the algorithm.  In this sense, our paper is “theoretical,” but we absolutely agree our paper is not in the field of Learning Theory, and we apologize if we gave this impression.  Nonetheless, we do believe the rigor of our motivation and derivation does provide new, interesting and principled ideas, and clearly is in a different class of methods from purely heuristic techniques, or the abundance of experimental papers that present good empirical results but only have conjectures as to why.
> > >
> > > 2. (“The presented “theory”…“) The key idea of our method, summarized in Algorithm 1, is showing a constructive approach whereby a ReLU network will maintain an exponential number of linear regions throughout initialization and training.  Theorem 3.1 ensures differentiability of the network output, which prevents fractal behavior of the network and further reduces error. The reviewer is correct that we do not have a theoretical result that says something to the effect of: if your network maintains an exponential number of linear regions, MSE loss will be exponentially (or some amount) lower.  For this, we point to, e.g., Montúfar et al. 2014 (“On the Number of Linear Regions of Deep Neural Networks”), Corollary 6, which tells that networks with more linear regions are able to compute more “complex” functions (functions that use more linear regions).  We propose to make a few citations like this in the Introduction section of our paper and quickly but clearly state up front in our paper that theory already exists that justifies that more linear regions yields more expressive ReLU networks.
> > >
> > > 3. (“I also don’t think…“) We suspect this may just be a subjective disagreement between the authors and the reviewer, which is okay.  However, we will at least offer a third-party opinion.  Hanin and Rolnick 2019 (which largely inspired our paper) write (we quote): “Our work suggests that realizing the full expressivity of deep networks may not be possible in practice, at least with current methods. […] A fundamental question in the theory of deep learning is why deeper networks often work better in practice than shallow ones. One proposed explanation is that, while even shallow neural networks are universal approximators, there are functions for which increased depth allows exponentially more efficient representations.” Thus, the Hanin and Rolnick paper invites a work like ours, which provides an algorithm for realizing the full expressivity of ReLU networks  - in preliminary cases, which we readily admit throughout our manuscript - and the fact that our algorithm is indeed possible adds a contribution to the ongoing discussion around this “fundamental question.”  Again, we are the first to admit that our present manuscript has clear limitations, and thus there is a lot more theoretical development to do - but we also invite the reviewers to consider the momentous consequences if we achieved a fully general version of this algorithm, and that there is still value in making progress towards this goal.  Nonetheless, we acknowledge that the value of an idea is a subjective judgment and do not want to belabor this point.

---

> > > ### Author Response · Authors · 2024-11-29
> > >
> > > 4. (“I don’t think…“) We are happy to take action on the concrete suggestion here. The reviewers have seen evidence of our ability to quickly add new experimental demonstrations of our method, on the two-spiral data set (a classification task, whereas before we only had regression examples), and on the concrete dataset in the UCI repository (an 8D example, where before we only had 1D and 2D).  We have begun working on experiments using CIFAR/ImageNet.  As with our other experiments, it will likely take us a couple of days to get these new results, so we will have to report the results textually in a follow-up comment after they’re run. We hope the reviewer can be understanding on this point. That being said, we can already make an educated guess as to the results: all the evidence in the paper, including the new examples we added in Appendix A.9, suggests that with higher-dimensional examples, our experimental network will marginally beat or perform comparably to a block-diagonal standard (Kaiming) network, but the dense variants will likely underperform a standard network, as the off-diagonal weights filled in by naive gradient descent won’t be used efficiently. It seems likely to be the case that on a per-parameter or per-time-complexity basis, the block diagonal algorithm variant can beat Kaiming-initialized dense layers. The remaining unknowns are whether some of the ingredients commonly present in medium-sized vision networks like convolution (we may need to use pretrained convolutional layers and test replacing just the linear layers), residuals, normalizations, data augmentations, or other hyperparameters somehow obscure the benefits our method provides.
> > >
> > > 5. (“Thanks for your…“) We will conduct evaluations on MNIST, CIFAR, and ImageNet as permitted by the remaining time.  If there are other specific datasets the reviewer wishes for us to try, please feel free to suggest them.  We do believe that the successes and limitations of the method are already illustrated - our lower-dimensional numerical experiments bear out the predictions of our current theory, and more theory is required to generalize the algorithm to achieve similar performance across all, arbitrarily deep and wide ReLU networks, and we state this and do not claim otherwise - but again, we remain receptive to specific suggestions as we’ve done with the other reviewers.
> > >
> > > 6. (“Explanation is indeed required…“) While you’re correct that most of the math (by page count) presented in this paper is about establishing that derivatives exist, etc., implicit in our construction is the usage of an exponential number of linear regions, and in that more limited sense the performance is proven. We believe the reviewer may be concerned that we aren’t explicit enough that having exponentially many linear regions yields a network that is “better” (more expressive).  As mentioned above, we will add a few sentences (with citations to theory) in our Introduction that make this fully clear from the outset.
> > >
> > > 7. (“Thanks for your…“) See point 1.
> > >
> > > We welcome further discussion on these (or other) points and will follow up shortly with the additional promised results.

---

> > > > ### Author Response · Authors · 2024-12-02
> > > >
> > > > As promised, we have shared results on CIFAR-10 and ImageNet - please see our general comment above on the page.  We also made the additional textual improvements to our paper’s introduction.  We believe we have now overcome all the objective criticisms presented by the reviewer.  While there are subjective differences that may remain, we are thankful for any further questions, comments, or suggestions the reviewer may have that may influence their view of our work, which has now undergone two rounds of substantial revisions and improvements since the initial submission.
> > > >
> > > > In re-reading the reviewer’s comments, we also want to stress one more time that we are not claiming our method is mathematically justified or the right thing to use for every ReLU network, e.g., this paper does *not* claim to displace Kaiming initialization or RAAI for every scenario.  Our initial examples in approximating convex and difference of convex functions are where we have justification for why we obtain an advantage, and we can obtain orders-of-magnitude better results in those cases. Examples like ImageNet (a modified VGG network), where more layers, convolutions, and other complexities are present cloud the analysis of what benefit to expect.  However, our intent was to make it very blunt in our manuscript that we do not claim or expect benefits on such large-scale and complex examples, and that future work is needed - our hope is that this is the first paper in an ongoing line of research.  We hope the limitations were clearly conveyed in the manuscript and that we did not give the impression that we were trying to claim too much; nonetheless, we thank the reviewer again for their insistence on running these large-scale tests, which surprisingly yielded further evidence in support of our method.

---

### Author Response · Authors · 2024-11-26

We thank the reviewers and ICLR staff again for their support of our work.  By taking a more mathematically principled approach, we’re able to train a small yet exponentially more efficient ReLU network, without needing to rely on batch norm, layer norm, dropout, or any other heuristics. Even though there is much more mathematical work needed to extend the results in our paper to networks of arbitrary depth and width, the ideas this paper contains are important to share with the community. We believe there is tremendous value in any learning publication - whether by us or any other group - that makes any progress towards exponentially more efficient neural networks.
Furthermore, we believe the reviewers’ constructive suggestions have substantially improved our work.  We now demonstrate higher-dimensional examples on real-world datasets, illustrating that our method already can be applied to practical learning tasks.  We recognize our revised manuscript as a substantial improvement over our initial submission are and grateful for this guidance.
Lastly, while we will not publicly remark on ongoing private conversations regarding ICLR policies and the reviews submitted by Reviewer EGry and Reviewer rFNb, we feel relatively confident the other two reviewers (or anyone in the public) can look at those reviews and hypothesize what our concerns are.  We humbly ask that you keep this in mind during the discussion period among reviewers and in your own evaluations of our work.

---

> ### Comment · Reviewer_EGry · 2024-11-27
>
> It's always great to see a new algorithm/method/approach designed, as long as strong evidence is presented to support its advantages with existing methods in either theoretical or empirical aspects, or ideally, in both. Unfortunately, this paper fails to present evidence in either.
>
> Theoretically, this paper proves the derivatives' existing conditions (for an indefinitely deep version), and discusses the monotonicity and continuity of the derivatives, with simple proofs. This is insufficient. The "theory" does not touch anything the learning theory community cares about (this is important because the author rebuttal claimed this is a theory paper). We don't have a clue about (1) whether the algorithm can converge and how fast it is, (2) the generalizability from training data to unseen test data, (3) the algorithmic stability against perturbation in input, etc.
>
> Empirically, this paper shows experiments in fitting one-dimensional data, fitting functions like $y = x^3 − x$ and $z = r^3$, and on a UCI dataset. It's far away from sufficiency, especially given no theoretical results on the algorithm's performance are presented. At least, empirical results on CIFAR-10/100, ImageNet (or a smaller variant of it), and an NLP dataset are required, given the authors claim the proposed method is generally for ReLU networks.
>
> If the authors do think their algorithm is good, please conduct theoretical analysis and/or empirical validation sufficiently. With strong evidence validating its performance, I believe their algorithm can be published in a top venue. Otherwise, I don't think this paper meet the threshold of ICLR.

---

> > ### Author Response · Authors · 2024-11-29
> >
> > We appreciate Reviewer EGry’s engagement and have addressed most of their comments below (under their review).  For instance, we acknowledge that we used the word “theory” in two different ways and do not claim to be a Learning Theory paper, which hopefully alleviates some (perhaps major) concerns.
> >
> > - “It’s always great…” We disagree that our paper “fails to prevent evidence” since (1) we show by construction how our algorithm can enable a ReLU network to maintain exponentially many linear regions, which is a novel mathematical development, and (2) we provide numerical experiments of our algorithm that verify that not only is this beneficial, but it can yield multiple orders-of-magnitude improvements in performance over other methods like Kaiming and RAAI.  The reviewer below recommends additional experiments, a point where we’re happy to oblige, but the reviewer’s characterization in this sentence is false with respect to the manuscript.
> >
> > - “Theoretically, this” We appreciate the reviewer viewed our proofs as simple - it was our goal to present the mathematics in the appendix in a readable way, so we are grateful we were successful in this regard.  On the rest of this comment, we again refer to some confusion on a “theoretical” paper vs. a Learning Theory paper, and we hope our discussion with the reviewer on this below alleviates most of the concern here.
> >
> > - “Empirically, this” We invite the reviewer to consider Section 5 of our manuscript, which also covers two-dimensional and non-convex functions (which is a non-obvious extension of our 1D approach).  It does appear the reviewer covered our new Appendix A.9, which evaluates noisy real-world data in higher dimensions.  While the characterization of “far away from sufficiency” feels like an overreach given what our paper is trying to claim, we are happy to report results from MNIST and/or CIFAR-10 (and/or ImageNet, if time permits).  We have detailed this in the reply to the reviewer, along with our timelines and expectations.  That being said, we are happy if the reviewer considers that a “far” improvement of our paper, since we expect to have results in a couple days.
> >
> > We have not found any other research paper presenting an algorithm like ours for maintaining exponentially many linear regions - an enticing upper bound that, as papers like Hanin and Rolnick 2019 point out, traditionally-trained ReLU networks generally fail to come anywhere near.  Yet other papers in the top venues have considered the possibility of such an algorithm and carefully explored region-counting and theoretical performance bounds.  A fully-general version of the present paper could, in the very best case, enable an *exponential*  increase in the approximation capability of ReLU networks, which would have an extraordinary impact on the field.  This motivated our excitement for the present work, and while we are one or several papers away from something that general (that would be quite a high bar), we believe that our algorithm represents an important advance that will inspire substantial future work towards that ultimate goal.  We look forward to reporting any additional results in the remainder of the discussion period and to continuing to discuss any points with the reviewers.

---

### Author Response · Authors · 2024-12-02
**Additional Experiments and Improvements**

We are pleased to share with the reviewers and AC additional experiments using our methodology.  Though PDF updates are no longer allowed, we have added these results to our manuscript as Appendix A.10, and have appropriately referenced the appendix from the main body.  (We have also added a few sentences (with citations) to our Introduction, as suggested by reviewer EGry, that emphasize that preserving more linear regions yields improved network expressivity - this was a straightforward improvement.)

Our additional experiments cover the two image classification problems suggested by reviewer EGry: CIFAR-10 and ImageNet.  We would like to highlight the drastic increase in scope and dimensionality of these examples (1000 image classes, convolutions, etc.) compared to our initial 1- and 2-D analytic function examples, and even the UCI and two-spiral examples we added per the suggestions of reviewers oevS and hKZ8.  Should evaluations on any further specific datasets be required, we are happy to have those suggested by reviewers as conditions of acceptance.

Here are the two figures of our new results, and the corresponding text we added to the manuscript:

Figure 16: https://postimg.cc/tn64y7WG

Figure 17: https://postimg.cc/1g4Psf17

We include very preliminary demonstrations of how our method can increase performance on large-scale image classification tasks. A nice feature of our method is that it can be used anywhere dense layers appear in networks, which enables its use in CNN architectures. Nonetheless, combining our construction with convolutions or other layers is a use case far exceeding the 1D test problems in the main body where we observe orders of magnitude of improvement, and the output of each subnetwork might not learn to be as nicely behaved as in the tightly controlled experimental setting of 1D convex training data. Thus, we stress that while our initial results are positive, and we can reasonably conjecture why, future work on extending our method to arbitrary ReLU networks would provide greater rigor and likely far greater performance gains, as seen with our lower-dimensional function approximation examples that realize the full benefit of our mathematically principled approach.

We evaluate two standard benchmarks: CIFAR-10 and ImageNet. In the case of CIFAR-10, we train a small network of 3 convolutions followed by 2 fully connected layers. To apply our technique, we insert our custom networks in between the two fully connected layers to act as activations, and reduce the number of neurons slightly to correspond with the extra parameters added. In the case of ImageNet, we use PyTorch’s pretrained VGG16 model for its convolution weights (which are frozen during training). We modify the linear layers at the end of the network in the same manner as we do for the CIFAR-10 experiment, and compare against re-learning the fully connected weights with Kaiming initialization. We are thus able to evaluate the effect of our network (on the linear layers) versus Kaiming initialization, for both CIFAR-10 and ImageNet.

Figure 16 shows the results for CIFAR-10. Our (“experimental”) training and validations generally remain below their Kaiming-initialized (“ordinary”) counterparts, including for instance at epochs 3 and 4, which per the loss curves’ characteristics are where one would likely stop training. While the loss curves are similar in shape, we report that at epoch 4, for example, our training and validation losses are 25.6% and 5.1% lower than their standard network counterparts.

In our ImageNet test (Figure 17), we similarly see modest gains of our (“experimental”) network over a Kaiming-initialized (“ordinary”) network, yet due to using a smaller learning rate here (0.00001, vs. 0.001 as used with CIFAR-10), the loss curves are smoother, so we see our networks consistently outperforming the standard network after 0.25 epochs (we trained for two full epochs but plot over the intermediate partial epochs). We trained both networks for two epochs; at epoch two, our training and validation losses were 0.01761 and 0.01657, compared to 0.02518 and 0.02269 for the standard networks (a 30% and 27% reduction, respectively).

These two examples show that even for large-scale image classification tasks - far away from the analytic 1D function tests considered in the main body - our methodology still has the potential to yield noticeable benefits, highlighting the promise of extending the present work to more general network architectures as a future research direction.

---

### Meta-Review · Area_Chair_mccE · 2024-12-23

**Metareview:**

This paper proposes a new training method that pretrains a ReLU network to a parameter having exponentially many linear regions before switching to naive gradient descent updates, which is demonstrated to have good approximation capabilities compared to standard initialization methods such as Kaiming initialization or RAAI.

This submission has gone through a lot of activity during the discussion period. Since the initial submission only contained 1-d/2-d experiments, all reviewers expressed concerns whether the proposed algorithm will be useful in practical high-dimensional datasets. The authors updated the paper with results on a 8-d UCI dataset and a 2-d two-spiral dataset (as well as other edits), and also presented some preliminary experiments on CIFAR-10 and ImageNet datasets (albeit trained only for a few epochs).

After the revision and the rebuttal, many reviewers updated their scores and the current average numerical score sits at the decision boundary. After a careful reading of the paper, reviews, and the discussion, my final evaluation is that the paper is unfortunately placed slightly below the bar for acceptance at ICLR.

The paper proposes a new training algorithm applicable to any linear layers, with the guarantee-by-design (at least during the pretraining stage) that the network parameters stay in the region with many “linear pieces” in the input space. I believe this can be a nice idea that has a good potential to develop into a new and efficient training procedure for ReLU networks in general. However, whenever one challenges the “de facto standard”, one needs to be equipped with strong theoretical guarantees on the performance and/or extensive experiments demonstrating the superiority of the proposed method. In my honest opinion, the submission at the moment fails to offer neither of them.

On the **theory** side, I think the presented theoretical results do not successfully demonstrate the advantage of the proposed algorithm. The theory is mostly on the characteristics (differentiability, concavity etc) of the function $F$ constructed by the infinite depth network. The network is guaranteed to have exponentially many linear regions in the input space up to the pretraining stage, but there is no theoretical guarantee after this stage: the training dynamics, convergence rates, generalization bounds, etc in the naive GD update stage are not considered. One can only intuitively expect that starting at a pretrained parameter with exponentially many number of linear piece should be helpful for a better approximation in the final stage.

In light of this, I must say that at least some of the authors’ claims regarding their theoretical guarantees are overly strong. For example,
- **“the theoretical results that explain why our method works are all proven”**: does not seem convincing from a learning theorist’s perspective.
-  **“theoretical guarantees are for convex functions and non-convex functions that are differences of convex functions”**: what kind of guarantees? In order to claim such a thing, we expect some universal approximation-type theorems. However, in the paper, I only see “any $F(x)$ with $s_i$ chosen according to Eq 3.1 is concave”, not something like “any convex function $g: [0,1] \to \mathbb{R}$ with $g(0) = 0$ can be approximated by $F$.”

On the **experiments** side, as all reviewers pointed out, in order to make significant impact, the proposed method must be tested on higher-dimensional real datasets. Although the authors put extensive efforts to address reviewers' concerns on the practical usefulness, I still wonder if the new results are significant.

In the new Appendix A.9, it seems to me that the “advantages” of the proposed algorithm over the ordinary training can only be seen when we restrict training to block diagonal parameters. The authors assert this is for fair comparison, but it's hard for me to see why we should restrict ourselves in a low-performing regime (block diagonal variants have higher test losses for both methods). Also, in order to really ensure fairness, I think the authors should have also considered memory/FLOP footprints and wall-clock time consumption.

The authors also reported preliminary experimental results on CIFAR-10 and ImageNet. Although the initial results look promising and I understand the lack of time, as an AC, I cannot make premature conclusions based on plots that only display 2 or 6 epochs of training. Also, some important training details are missing here—whether dense or block diagonal version is used, how many epochs correspond to the pretraining stage (as “switch-off” happens at 500 epochs in Fig 14), why can CIFAR-10 losses be such small after only 6 epochs, etc. Such missing details prevent me from accurately assessing the validity and significance of the preliminary results.

Overall, I think the paper has clear merits but the results are not ripe enough to be presented at ICLR. At this time, I recommend rejection.

**Additional Comments On Reviewer Discussion:**

Based on other comments from the discussion,
- I think it would be helpful to elaborate a little more on how backpropagation is done under the reparameterization in the next version.
- It would be challenging but nice to prove some theorems on the training trajectory after pretraining, to strengthen the theoretical guarantees.

---

### Decision · Program_Chairs · 2025-01-22

Reject